# Rare-Earth Doped $Gd_{3-x}RE_xFe_5O_{12}$ (RE = Y, Nd, Sm, and Dy) Garnet: Structural, Magnetic, Magnetocaloric, and DFT Study

Dipesh Neupane [1], Noah Kramer [2], Romakanta Bhattarai [1], Christopher Hanley [2], Arjun K. Pathak [2], Xiao Shen [1], Sunil Karna [3] and Sanjay R. Mishra [1,*]

[1] Department of Physics and Materials Science, The University of Memphis, Memphis, TN 38152, USA; dipeshneupane07@gmail.com (D.N.); bhattr2@rpi.edu (R.B.); xshen1@memphis.edu (X.S.)
[2] Department of Physics, SUNY Buffalo State, Buffalo, NY 14222, USA; kramern01@mail.buffalostate.edu (N.K.); hanleycm01@mail.buffalostate.edu (C.H.); pathakak@buffalostate.edu (A.K.P.)
[3] Department of Health and Natural Science, Union College, Barbourville, KY 40906, USA; skarna@unionky.edu
* Correspondence: srmishra@memphis.edu

**Abstract:** The study reports the influence of rare-earth ion doping on the structural, magnetic, and magnetocaloric properties of ferrimagnetic $Gd_{3-x}RE_xFe_5O_{12}$ (RE = Y, Nd, Sm, and Dy, $x$ = 0.0, 0.25, 0.50, and 0.75) garnet compound prepared via facile autocombustion method followed by annealing in air. X-Ray diffraction (XRD) data analysis confirmed the presence of a single-phase garnet. The compound's lattice parameters and cell volume varied according to differences in ionic radii of the doped rare-earth ions. The $RE^{3+}$ substitution changed the site-to-site bond lengths and bond angles, affecting the magnetic interaction between site ions. Magnetization measurements for all $RE^{3+}$-doped samples demonstrated paramagnetic behavior at room temperature and soft-ferrimagnetic behavior at 5 K. The isothermal magnetic entropy changes ($-\Delta S_M$) were derived from the magnetic isotherm curves, $M$ vs. $T$, in a field up to 3 T in the $Gd_{3-x}RE_xFe_5O_{12}$ sample. The maximum magnetic entropy change ($-\Delta S_M^{max}$) increased with $Dy^{3+}$ and $Sm^{3+}$ substitution and decreased for $Nd^{3+}$ and $Y^{3+}$ substitution with $x$ content. The $Dy^{3+}$-doped $Gd_{2.25}Dy_{0.75}Fe_5O_{12}$ sample showed $-\Delta S_M^{max} \sim 2.03$ $Jkg^{-1}K^{-1}$, which is ~7% higher than that of $Gd_3Fe_5O_{12}$ (1.91 $Jkg^{-1}K^{-1}$). A first-principal density function theory (DFT) technique was used to shed light on observed properties. The study shows that the magnetic moments of the doped rare-earths ions play a vital role in tuning the magnetocaloric properties of the garnet compound.

**Keywords:** garnet; rare-earth doped garnet; X-ray diffraction; magnetic; magnetocaloric; DFT

## 1. Introduction

Magnetic refrigeration (MR) technology based on the magnetocaloric effect (MCE) principle has been considered a promising alternative to replace conventional vapor compression cooling technology [1–4]. It is an intrinsic magneto-thermal response of magnetic materials [5]. Materials exhibit MCE by inducting adiabatic heating or cooling in the applied magnetic field. One of the quantitative parameters to characterize magnetocaloric materials (MCM) is the isothermal magnetic entropy change ($\Delta S_M$), which is induced by a change in an applied magnetic field ($\Delta H$) [1]. Recently, a Gd-based garnet, $Gd_3Fe_5O_{12}$, has received attention due to its high MCE at low temperatures (below 50 K), making it suitable for liquefaction processes, cryogenic technology, and space applications [2–4]. $Gd_3Fe_5O_{12}$ belongs to an essential class of iron garnet materials due to their significant magnetocaloric [6], magneto-optic [7,8], recording device [9], microwave device [10], sensing [11], and magnetic properties [10,11].

$Gd_3Fe_5O_{12}$ is a complex ceramic oxide, having the chemical formula $A_3B_2C_3O_{12}$ (where A = $RE^{3+}$ ion, B and C = $Fe^{3+}$ ions). The unique crystal symmetry of a garnet plays a crucial role in its physical properties. The garnet structure holds a wide variety of cations. The structure consists of three different crystallographic sites, namely, dodecahedral (c),

octahedral (a), and tetrahedral (d), where 24A ions reside in the (c) site, 16B ions in the (a) site, and 24C ions in the (d) site. The unit cell of the garnet structure contains eight formula units of ${Gd_3}[Fe1_2](Fe2_3)O_{12}$ arranged as a framework of metal-oxygen polyhedra formed from (a) and (d) site cations. Here, { }, [ ], and ( ) represent the three different cationic sublattices. These cations are located at the centers of the corresponding polyhedrons, as shown in Figure 1. $Gd^{3+}$ ions occupy the dodecahedral (Figure 1a) site with position 24c, while Fe1 and Fe2 ions occupy octahedral and tetrahedral sites with positions 16a and 24d (Figure 1b,c). The arrangements of different polyhedral and oxygen ions are given in Figure 2a for the garnet structure. The crystal structure of $Gd_3Fe_5O_{12}$ with eight formula units per cell is shown in Figure 2b.

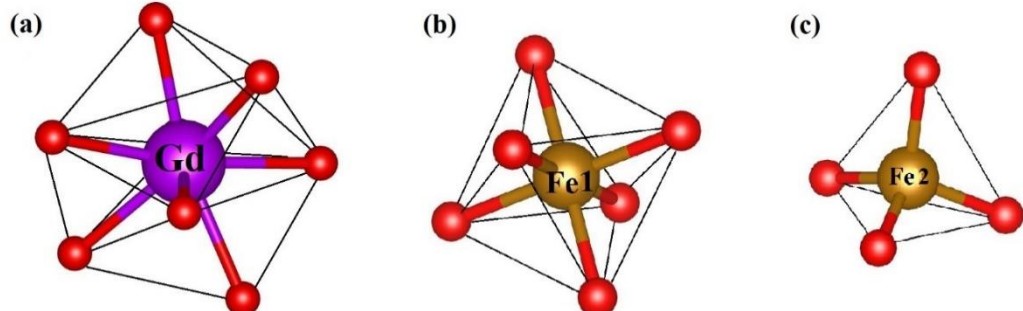

**Figure 1.** Different polyhedral arrangements of cations in $Gd_3Fe_5O_{12}$ (**a**) dodecahedral, (**b**) octahedral, and (**c**) tetrahedral.

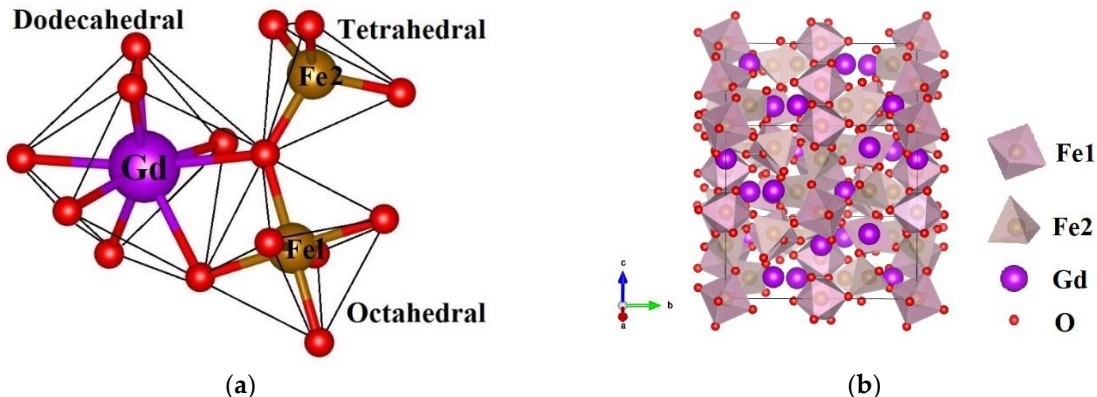

**Figure 2.** (**a**) The arrangement of different polyhedral cells in a unit cell of $Gd_3Fe_5O_{12}$. (**b**) The crystal structure of $Gd_3Fe_5O_{12}$ with eight formula units per unit cell.

In $Gd_3Fe_5O_{12}$, two sub-lattices of ferric ions couple anti-ferromagnetically in the superexchange interaction via oxygen anions. The formula unit consists of three $Fe^{3+}$ cations on tetrahedral sites and two $Fe^{3+}$ cations on octahedral sites. The $Gd^{3+}$ ions are also antiferromagnetically coupled to the net moment of the $Fe^{3+}$ ions, but this coupling is weaker than that between $Fe^{3+}$ ions. Since the $Gd^{3+}$ ions are disordered at room temperature, the ferri-magnetic properties of the material at high temperatures are governed by the moments of the $Fe^{3+}$ ions [12–14]. It is known that $Fe^{3+}$ ions at octahedral and tetrahedral sites provide a positive and negative contribution to the compound's net magnetic moment. At low temperatures, however, the $Gd^{3+}$ lattice becomes ordered and dominates the material's magnetic properties due to the more significant magnetic moment of $Gd^{3+}$ (below 90 K [3]) compared with $Fe^{3+}$ ions. The magnetic and magnetocaloric properties largely depend on the total angular momentum quantum number (*J*). The increase in the *J* value is expected to increase the magnetic moment of each magnetic cluster in the garnet and lead to a rise

in the $\Delta S_M$ value. The bulk garnet magnetization as a function of temperature can be written as

$$M(T) = M_c(T) - [M_d(T) - M_a(T)]$$

where $M_c(T)$ is the magnetization of the $Gd^{3+}$ sublattice, and $M_d(T)$ and $M_a(T)$ are the magnetizations of $Fe^{3+}$ at tetrahedral and octahedral sublattices, respectively. At low temperatures, the magnetization of $RE^{3+}$ sublattices is more than that of the $Fe^{3+}$ sublattices. As $T$ increases, the magnetization of the $RE^{3+}$ sublattices decreases faster than $Fe^{3+}$ sublattices and reaches a point where the net moment is zero. The temperature at this point is called the compensation temperature ($T_{comp}$). Above the compensation temperature, the net magnetization of the iron sublattices $[M_d(T) - M_a(T)]$ exceeds that of the $RE^{3+}$ sublattice, resulting in a rise in the magnetization [15]. This is because the rare-earth and iron sublattice moments randomize at different temperatures. For example, $Ho_3Fe_5O_{12}$ shows $T_{comp}{\sim}127$ K [16], and $Er_3Fe_5O_{12}$ shows $T_{comp}$ at 186 K [17].

The $Gd_3Fe_5O_{12}$ compound displays high magnetocaloric properties at low temperatures associated with intrinsic magnetic frustration and magnetic ordering of the $Gd^{3+}$ sublattice [3]. The intrinsic magnetic properties of the garnet are affected by the partial substitution for $Gd^{3+}$ or $Fe^{3+}$ sites or both. Nguyet et al. studied the crystallization and magnetic characterization of $(Dy, Ho)_3Fe_5O_{12}$ nanopowders prepared using a sol-gel technique [18]. They reported a sizeable magnetic susceptibility and coercivity compared to the corresponding values for bulk samples, a trend attributed to the disordered nature of the surface spin of single-domain particles. Jie et al. studied the structural and magnetic properties of Ca- and Sr-doped $Nd_3Fe_5O_{12}$ nanopowders prepared using a hydrothermal method [19]. The particle size of $Nd_{3-x}(Ca, Sr)_xFe_5O_{12}$ decreased with the concentration of Ca and Sr, while the saturation magnetization value decreased due to the weak exchange interaction. Li et al. studied the MCE in heavy rare-earth iron garnets ($Ho_3Fe_5O_{12}$ and $Er_3Fe_5O_{12}$) [20]. $Ho_3Fe_5O_{12}$ and $Er_3Fe_5O_{12}$ displayed a compensation effect characterized by a zero magnetization at 134 K and 80 K, respectively. The reported maximum magnetic entropy change value at the 5 T field is 4.72 $Jkg^{-1}K^{-1}$ for $Ho_3Fe_5O_{12}$ at 34 K and 4.94 $Jkg^{-1}K^{-1}$ for $Er_3Fe_5O_{12}$ at 24 K, respectively. Aparnadevi et al. studied the structural and magnetic behavior of Bi-doped $Gd_3Fe_5O_{12}$ prototype garnet synthesized via the ball milling method [21]. A shift in the Curie point towards the high-temperature region was observed and ascribed to the stabilizing effect of Bi ion on magnetic ordering. Canglong Li et al. studied the magnetocaloric effect in $RE_3Fe_5O_{12}$ (RE = Gd, Dy) synthesized using a sol-gel method [22]. The maximum value of $-\Delta S_M$ achieved 3.40 $Jkg^{-1}K^{-1}$ at 40 K and 3.51 $Jkg^{-1}K^{-1}$ at 58 K, for RE = Gd and Dy, respectively, reflecting the influence of the difference in magnetic moments of $Gd^{3+}$ and $Dy^{3+}$.

The ionic radii of these rare-earth ions are $Dy^{3+}{\sim}0.912$ Å, $Nd^{3+}{\sim}0.983$ Å, $Sm^{3+}{\sim}0.958$ Å, and $Y^{3+}{\sim}0.90$ Å [23], and their corresponding magnetic moments are 10 $\mu_B$ for $Dy^{3+}$ [24], 1.14 $\mu_B$ for $Nd^{3+}$ [25], 0.74 $\mu_B$ for $Sm^{3+}$ [26], and 0 for $Y^{3+}$ [27]. Considering these ionic radii and magnetic moment trends, the rare-earth substitution in Gd garnet is expected to bring a new magnetic order in the compound. A detailed study of the $RE^{3+}$ doping effect on the structural, magnetic, and magnetocaloric properties of $Gd_3Fe_5O_{12}$ garnet is lacking. Suitable $RE^{3+}$ substitution in $Gd_3Fe_5O_{12}$ is expected to bring changes in the lattice structure, magnetic moment, and exchange-coupling, affecting the compound's magnetic and magnetocaloric properties. The present work reports a detailed study on the effect of $RE^{3+}$ substitution in $Gd_{3-x}RE_xFe_5O_{12}$, ($RE^{3+}$ = Y, Nd, Sm, and Dy, $x$ = 0.0, 0.25, 0.50, and 0.75) garnet compound. For example, in $Gd^{3+}$-rich $Gd_3Fe_5O_{12}$, the $Gd^{3+}$ ion is an $^8S_{7/2}$-state (J = 7/2, L = 0), and the magnetic moment per ion is 7 $\mu_B$. Thus, the $Gd^{3+}$ ion is not affected by the crystalline field. The system is isotropic, with the $Gd^{3+}$ moment following the applied magnetic field. Therefore, it is easy to align other substituted anisotropic ions towards the hard direction of magnetization when $Gd^{3+}$ is replaced with other rare-earth ions in a small amount. However, in $RE_3Fe_5O_{12}$ compounds other than Gd, with RE = $Ho^{3+}$ (a non-S state ion), the crystalline electric field causes quenching of the orbital angular momentum L. The exchange field plus the crystal electric field will cause the $RE^{3+}$ moments

to assume a conical arrangement relative to the easy magnetization direction, which is also possible [28–30].

In the present work, we report the results of detailed structural, magnetic, magnetocaloric, and Mossbauer spectral studies of rare-earth ion substituted $Gd_{3-x}RE_xFe_5O_{12}$ ($RE^{3+}$ = Y, Nd, Sm, and Dy) garnet, with the compounds being synthesized using an autocombustion technique. The chosen rare-earth ions $Y^{3+}$ (zero magnetic moments), $Sm^{3+}$ and $Dy^{3+}$ (positive magnetic moment), and $Nd^{3+}$ (negative moment) are expected to have a marked influence on the exchange-interaction and dipole–dipole interaction in the ferrimagnetic $Gd_{3-x}RE_xFe_5O_{12}$ compound.

## 2. Experimental Details

### 2.1. Synthesis

$Gd_{3-x}RE_xFe_5O_{12}$, $RE^{3+}$ = Y, Nd, Sm, and Dy, $x$ = 0.0, 0.25, 0.50, and 0.75 samples were synthesized via an autocombustion method using glycerin as a chemical reagent. Nitrate salts of rare-earth, $Gd(NO_3)_3 \cdot 6H_2O$ and $Fe(NO3)_3 \cdot 9H_2O$, were mixed in the stoichiometric amount into deionized water. Glycine-to-metal nitrate molar ratios of 1:8 were mixed as a combustion reagent fuel. The solution was ultrasonicated for 60 min. Glycine complexes the metal cations, thereby preventing selective precipitation and oxidizing by nitrate anions, thereby serving as a fuel for combustion [31]. The mixture was heated to 80 °C until a brown viscous gel formed. Instantaneously, the gel ignited, forming copious amounts of gas, resulting in a lightweight, voluminous powder. The resulting "precursor" powder was calcined at 1100 °C for 12 h to obtain pure $RE^{3+}$ doped $Gd_{3-x}RE_xFe_5O_{12}$ iron garnet. Table 1 provides the stoichiometry of the chemicals used in the synthesis.

**Table 1.** Stoichiometry of chemicals used in the synthesis of the $Gd_{3-x}RE_xFe_5O_{12}$ compound.

| $Gd_{3-x}RE_xFe_5O_{12}$ $RE^{3+}$ | $x$ | $Gd(NO_3)_3 \cdot 6H_2O$ | $Fe(NO_3)_3 \cdot 9H_2O$ | $Dy(NO_3)_3 \cdot H_2O$ | $Nd(NO_3)_3 \cdot 6H_2O$ | $Sm(NO_3)_3 \cdot 6H_2O$ | $Y(NO_3)_3 \cdot 5H_2O$ | Glycine |
|---|---|---|---|---|---|---|---|---|
| | | | | Weight in gm. | | | | |
| Dy | 0.00 | 0.718 | 1.071 | 0.000 | - | - | - | 0.318 |
| | 0.25 | 0.657 | 1.069 | 0.046 | - | - | - | 0.318 |
| | 0.50 | 0.596 | 1.068 | 0.092 | - | - | - | 0.317 |
| | 0.75 | 0.536 | 1.066 | 0.138 | - | - | - | 0.317 |
| Nd | 0.00 | 0.718 | 1.071 | - | 0.000 | - | - | 0.318 |
| | 0.25 | 0.661 | 1.075 | - | 0.058 | - | - | 0.319 |
| | 0.50 | 0.602 | 1.078 | - | 0.117 | - | - | 0.321 |
| | 0.75 | 0.544 | 1.082 | - | 0.176 | - | - | 0.321 |
| Sm | 0.00 | 0.718 | 1.071 | - | - | 0.000 | - | 0.318 |
| | 0.25 | 0.659 | 1.073 | - | - | 0.059 | - | 0.319 |
| | 0.50 | 0.601 | 1.075 | - | - | 0.118 | - | 0.319 |
| | 0.75 | 0.541 | 1.077 | - | - | 0.177 | - | 0.321 |
| Y | 0.00 | 0.718 | 1.071 | - | - | - | 0.000 | 0.318 |
| | 0.25 | 0.671 | 1.091 | - | - | - | 0.049 | 0.324 |
| | 0.50 | 0.621 | 1.111 | - | - | - | 0.100 | 0.33 |
| | 0.75 | 0.569 | 1.132 | - | - | - | 0.153 | 0.336 |

## 2.2. Characterization

X-ray diffraction (XRD) experiment was conducted with $CuK\alpha_1$ ($\lambda \sim 1.5406$ Å) radiation using a D8 Advance diffractometer (Bruker, Germany) to examine the phase purity and structural characteristics of the prepared sample. The powder X-ray data were collected in the $2\theta$ range from $20°$ to $70°$ with a step size of $0.042°$ and collection time of 0.2 s/step using a solid-state Vantec detector (Bruker). The morphology of the samples was analyzed using scanning electron microscopy, SEM (Phenom model number PW-100-015 at 10 kV. The magnetic properties of samples viz. hysteresis and field-cooled (FC) and zero-filed-cooled (ZFC) measurements were performed using a physical property measurement system (PPMS, Quantum Design, San Diego, CA, USA) as a function of temperature in the range 5–300 K and field up to 3 T. The sample was cooled down to 5 K for the ZFC measurement without an external field. Then, the magnetic field of 100 Oe was applied to the system, followed by magnetization measurement as a function of temperature from 5 K to 300 K. The FC measurement was performed by lowering the system temperature to 5 K in a 100 Oe field. The FC magnetization as a function of temperature was recorded in warming-up conditions from 5 to 300 K. To calculate magnetic entropy change, isothermal magnetization curves were collected in a field up to 3 T in a temperature step of 7 K.

## 2.3. Density Functional Theory

First-principles density functional theory (DFT) calculations were performed for the self-consistent calculations and geometry optimizations. The DFT+U method [32] was used with the Perdew–Burke–Ernzerhof (PBE) [33] version of the exchange-correlation functional. The calculations were performed using the Vienna ab initio simulation package (VASP) [34] under the projected-augmented wave (PAW) [35] pseudo-potential. The structure $Gd_{3-x}RE_xFe_5O_{12}$ considered in the calculation has a size of 80 atoms in total. Respective structures with the variable $x$ ranging from 0.0 to 1.0 along with a step of 0.25, were considered in the calculations. All the data is taken from the relaxed structures after optimization. Note that in the Dudarev approach of DFT+U [32], the parameters U and J do not enter calculations separately; instead, an effective Coulomb-exchange interaction $U_{eff} = U - J$ is used.

## 3. Results and Discussion

### 3.1. Structural Properties

#### 3.1.1. Phase Analysis

The XRD was performed on a powder sample and finally spread on a zero background sample holder. Figure 3 shows the room temperature XRD pattern of $Gd_{3-x}RE_xFe_5O_{12}$ (RE = Y, Nd, Sm, and Dy, $x$ = 0.0, 0.25, 0.50, and 0.75). Single-phase garnet structure (ICDD card no. 01-083-1027) with a cubic crystalline phase group $Ia\bar{3}d$ was evident for $x < 0.75$. An impurity, $GdFeO_3$, appears at higher doping content, $x$ = 0.75, for $Nd^{3+}$, $Sm^{3+}$, and $Y^{3+}$. The $RE^{3+}$ substitution shows a gradual shift in the XRD peaks compared to pure $Gd_3Fe_5O_{12}$ (inset Figure 3). The observed shifts are in accordance with the difference in ionic radii of substituted $RE^{3+}$ compared to $Gd^{3+}$ ($r \sim 0.938$ Å) in octahedral symmetry where $Dy^{3+}$ ($r$ = 0.912 Å) and $Y^{3+}$ ($r$ = 0.90 Å) ions are smaller, and $Nd^{3+}$ ($r$ = 0.983 Å) and $Sm^{3+}$ ($r$ = 0.958 Å) are bigger than $Gd^{3+}$ ion [23].

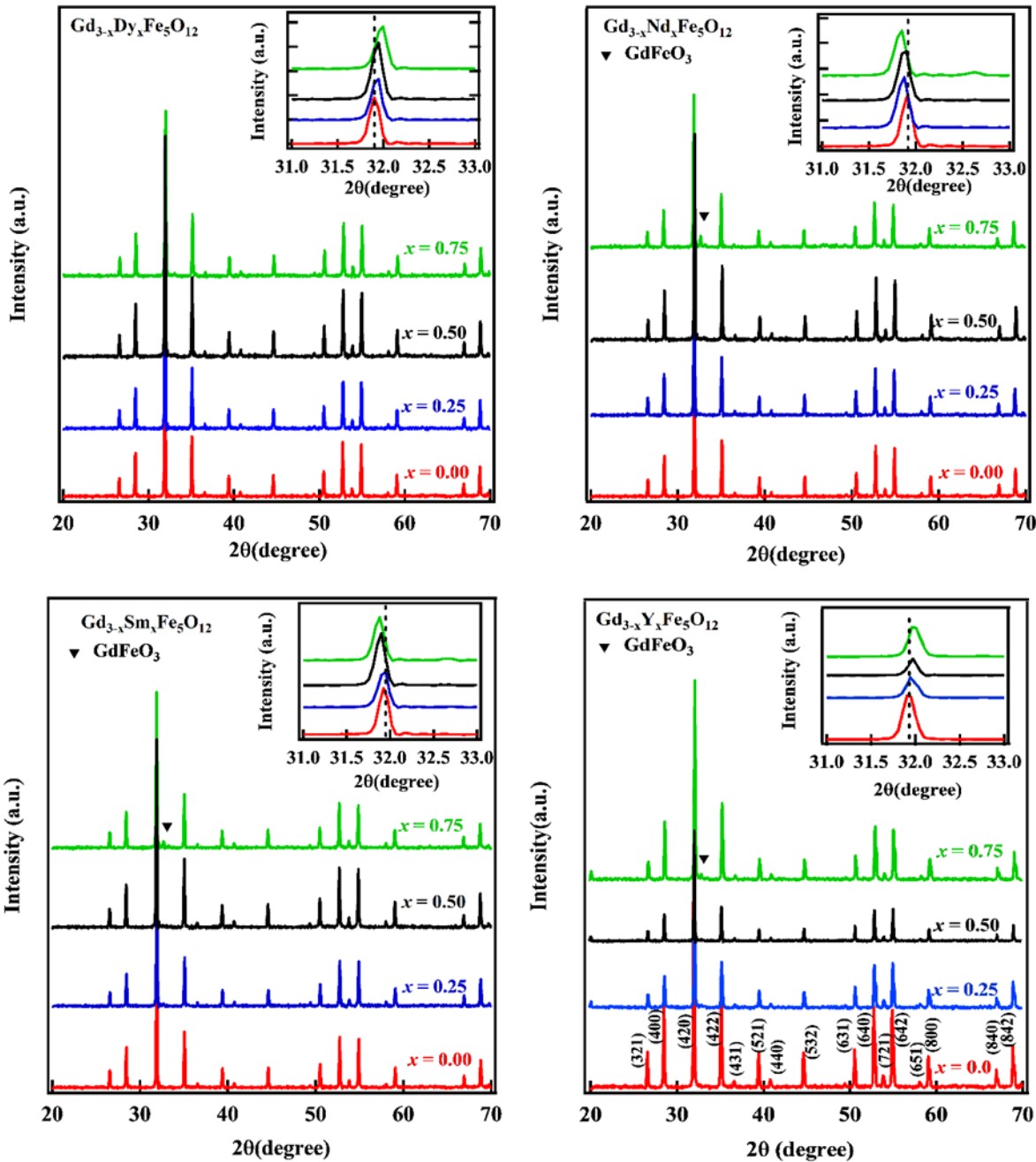

**Figure 3.** XRD pattern of the $Gd_{3-x}RE_xFe_5O_{12}$ compound. The inset shows an expanded view of the XRD pattern between 31–33°.

The structural analysis was carried out via the Rietveld [36] refinement technique using GSAS [37] software. The fitted powder profile of $Gd_{3-x}RE_xFe_5O_{12}$ is presented in Figures 4–7, and the structural parameters derived from Rietveld refinement are listed in Table 2.

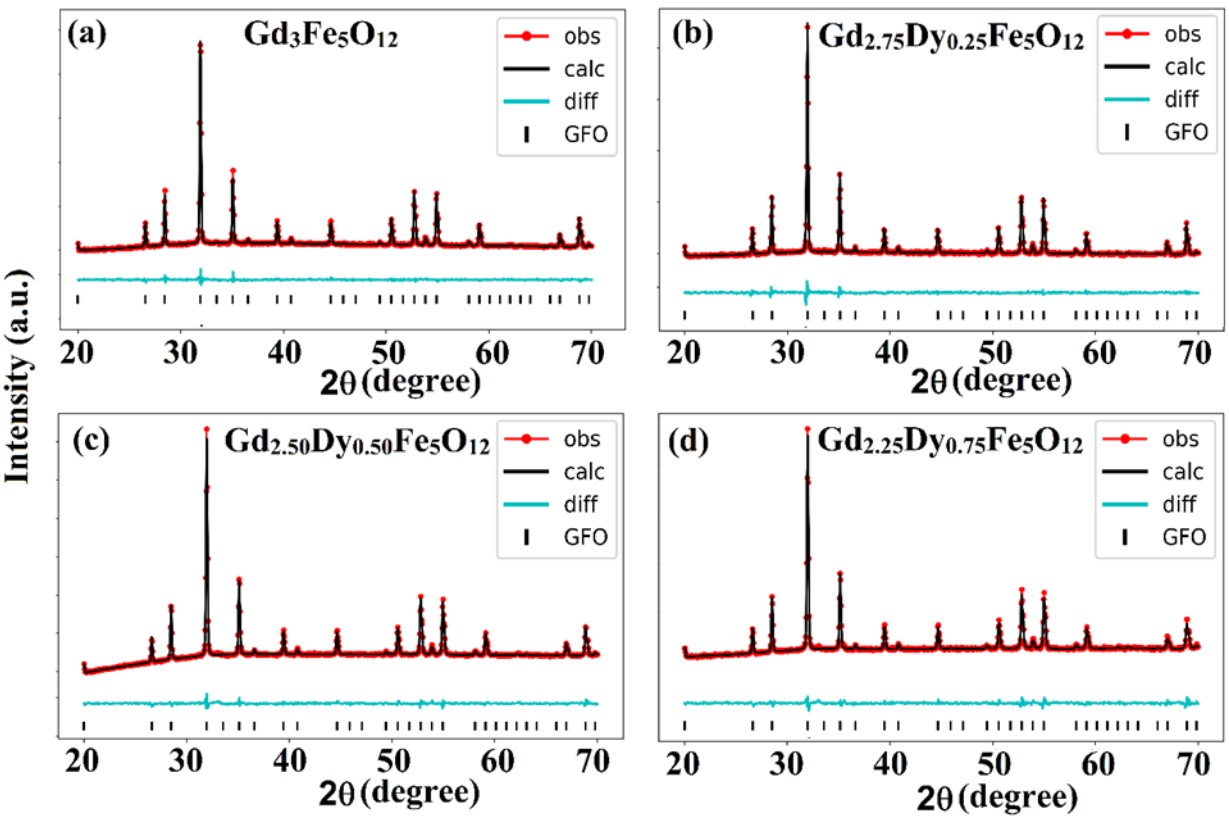

**Figure 4.** (**a**–**d**): Rietveld refinement profile for $Gd_{3-x}Dy_xFe_5O_{12}$ compound.

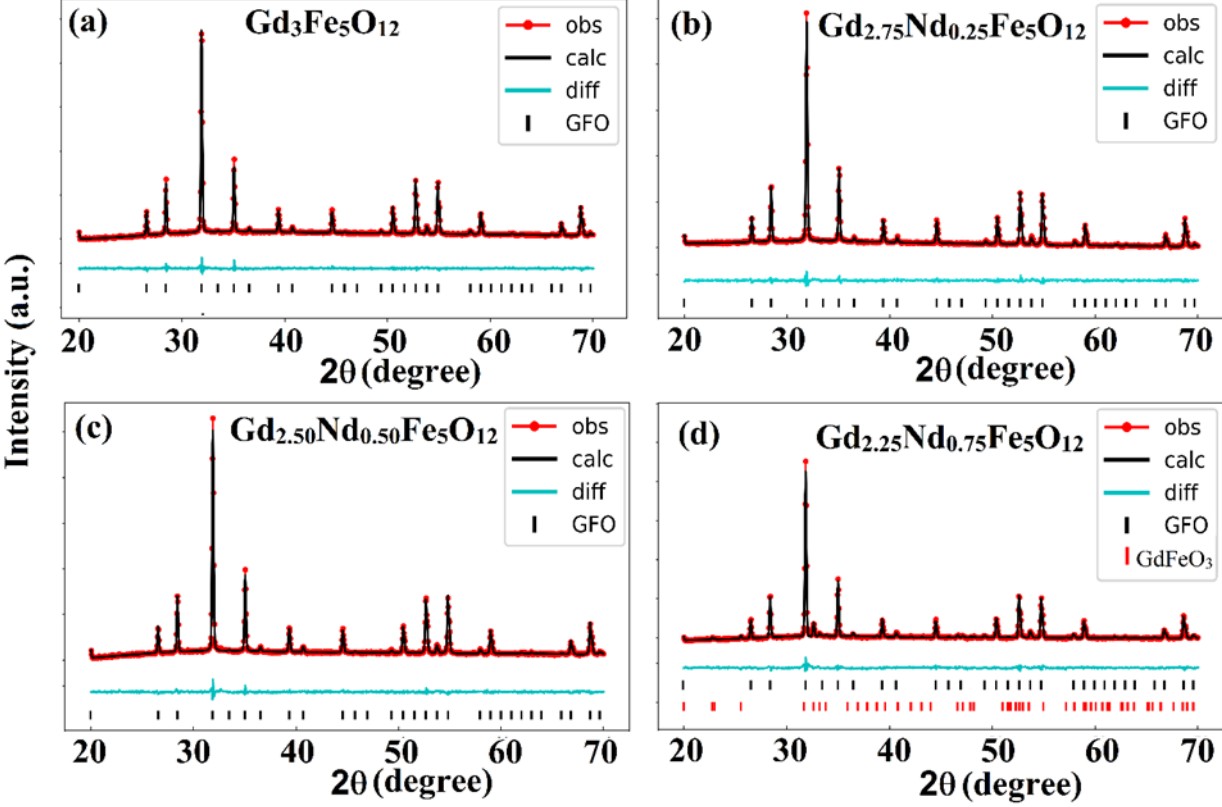

**Figure 5.** (**a**–**d**): Rietveld refinement profile for the $Gd_{3-x}Nd_xFe_5O_{12}$ compounds.

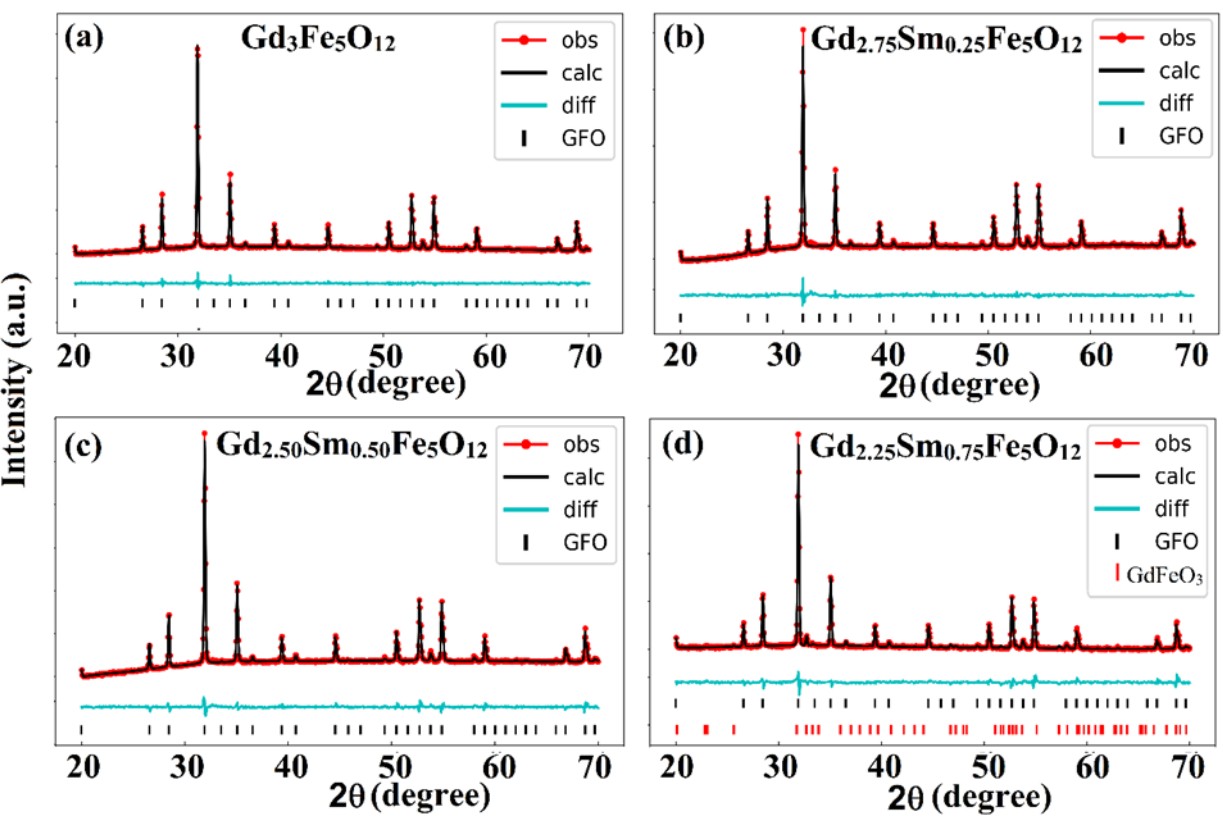

**Figure 6.** (**a**–**d**): Rietveld refinement profile for the $Gd_{3-x}Sm_xFe_5O_{12}$ compounds.

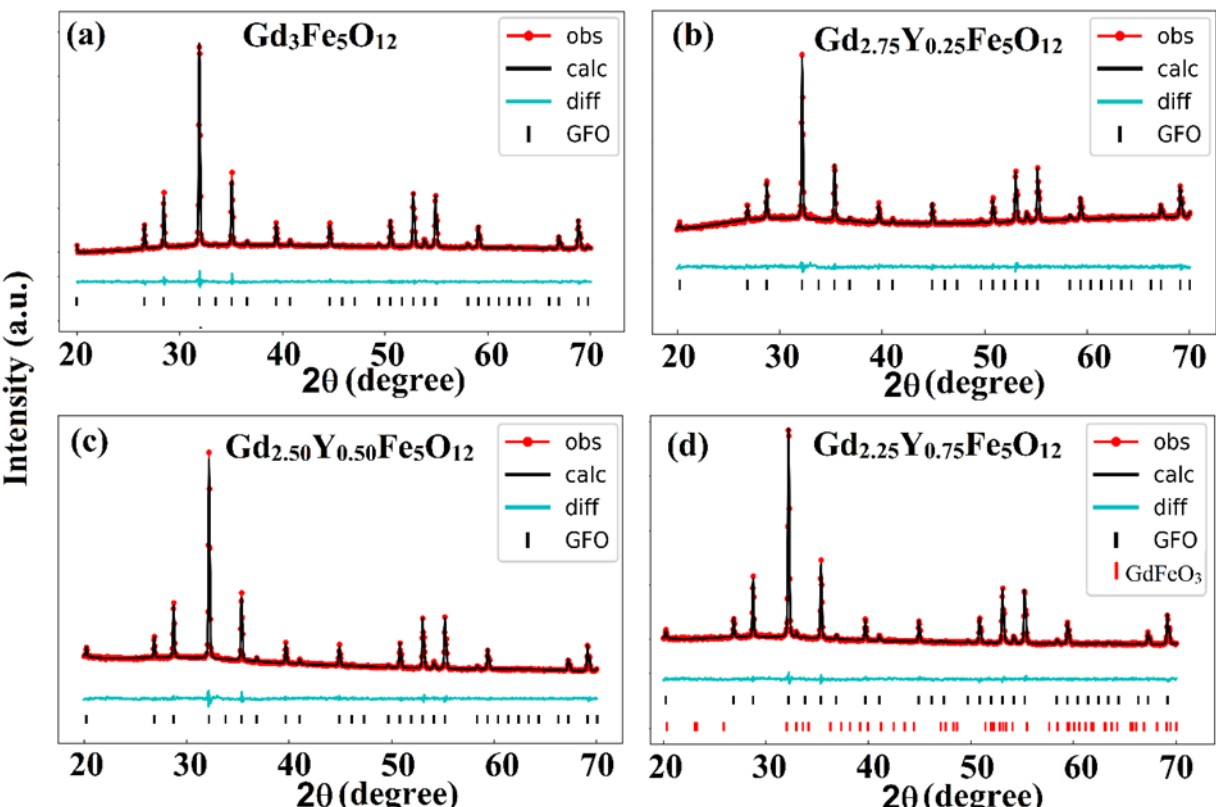

**Figure 7.** (**a**–**d**): Rietveld refinement profile for the $Gd_{3-x}Y_xFe_5O_{12}$ compounds.

**Table 2.** Structural parameters derived from Rietveld refinement of powder XRD data of the $Gd_{3-x}RE_xFe_5O_{12}$ compound.

| $Gd_{3-x}RE_xFe_5O_{12}$ | $x$ | $a$ (Å) | $V$ (Å$^3$) | O(x) | O(y) | O(z) | Density (g/cm$^3$) | $R_{wp}$ (%) | $\chi^2$ |
|---|---|---|---|---|---|---|---|---|---|
| Dy | 0.00 | 12.4693(11) | 1938.629(5) | −0.0296 | 0.0538 | 0.1467 | 6.487 | 1.194 | 1.64 |
| | 0.25 | 12.4676(14) | 1937.697(6) | −0.0283 | 0.0562 | 0.1492 | 6.489 | 1.622 | 2.82 |
| | 0.50 | 12.4647(12) | 1936.299(5) | −0.0297 | 0.0563 | 0.1470 | 6.485 | 1.351 | 2.55 |
| | 0.75 | 12.4616(13) | 1935.367(18) | −0.0279 | 0.0561 | 0.1496 | 6.497 | 1.515 | 4.21 |
| Nd | 0.00 | 12.4693(11) | 1938.629(5) | −0.0296 | 0.0538 | 0.1467 | 6.487 | 1.194 | 1.64 |
| | 0.25 | 12.482213) | 1944.699(6) | −0.0292 | 0.0524 | 0.1483 | 6.419 | 1.269 | 1.88 |
| | 0.50 | 12.4899(12) | 1948.422(5) | −0.0306 | 0.0545 | 0.1472 | 6.430 | 1.207 | 1.80 |
| | 0.75 | 12.5018(13) | 1953.594(6) | −0.0273 | 0.0567 | 0.1449 | 6.346 | 1.575 | 2.67 |
| Sm | 0.00 | 12.4693(11) | 1938.629(5) | −0.0296 | 0.0538 | 0.1467 | 6.487 | 1.194 | 1.64 |
| | 0.25 | 12.4746(13) | 1940.963(6) | −0.0282 | 0.0558 | 0.1481 | 6.474 | 1.249 | 1.97 |
| | 0.50 | 12.4848(14) | 1946.077(7) | −0.0298 | 0.0554 | 0.1461 | 6.423 | 1.586 | 3.50 |
| | 0.75 | 12.4925(19) | 1949.377(9) | −0.0307 | 0.0573 | 0.1492 | 6.480 | 1.982 | 4.38 |
| Y | 0.00 | 12.4693(11) | 1938.629(5) | −0.0296 | 0.0538 | 0.1467 | 6.487 | 1.194 | 1.64 |
| | 0.25 | 12.4648(13) | 1936.298(6) | −0.0284 | 0.0558 | 0.1460 | 6.392 | 1.406 | 1.72 |
| | 0.50 | 12.4550(11) | 1932.136(5) | −0.0295 | 0.0537 | 0.1481 | 6.235 | 1.458 | 2.24 |
| | 0.75 | 12.4466(32) | 1928.386(2) | −0.026 | 0.0580 | 0.1460 | 6.100 | 1.256 | 1.64 |

### 3.1.2. Bond Angle and Bond Length

The Rietveld refinement reveals a good match (R-weighted parameter ($R_{wp\,\%}$) < 2.0%) of the observed and calculated profiles for all the samples. The lattice parameters, density, oxygen coordinate, and Rwp (%) obtained from the Rietveld refinement are listed in Table 2. It can be seen that the value of the lattice parameter $a$ and unit cell volume $V$ changes with the RE$^{3+}$ content $x$. The changes in the lattice parameter with $x$ obey Vegard's law [38].

Each of the three positive ion positions in the garnet structure is associated with a different coordination polyhedron of oxygen ion. In $Gd_3Fe_5O_{12}$, Fe$^{3+}$ Octa (Figure 8a), Fe$^{3+}$ tetra (Figure 8b), and Gd$^{3+}$ dodeca (Figure 8c) have a regular polyhedral with respect to edge length. The atom-to-atom angles (Gd–O–Fe1(Oct.), Gd–O–Fe2(Tetra.), and Fe1–O–Fe2 and site-to-site bond distances (Gd–Fe1, Gd–Fe2, Fe1–O, Fe2–O, and Fe1–Fe2) are listed in Table 3 and plotted in Figure 9a,b, Figure 10a,b, respectively. The bond lengths Gd–Fe1 decreased with the Dy$^{3+}$ and Y$^{3+}$ substitution, whereas Nd3+ and Sm3+ substitution increased. The bond length of RE–Fe1 is similar to the bond length calculated by S. Geller et al. for the $Y_3Fe_5O_{12}$ garnet [39], as listed in Table 3. Due to the high centrosymmetric nature of the cubic garnet structure, the bond angles, such as Fe1–Gd–Fe2 (56.8°) and Gd–O–Gd, etc., remain largely unaltered. In contrast, the bond angle Fe1–O–Fe2 decreases, and the bond angle Gd–O–Fe1 increases with RE$^{3+}$ substitution following the ionic size difference between the Gd$^{3+}$ and RE$^{3+}$ ions. The magnetic interactions between ions largely depend on bond-angle and bond-length values. Magnetic interactions cannot occur via the conduction of electrons in garnet due to their insulating nature. The magnetic ions of garnet are separated from each other by the large oxygen ions, and this separation is too large to give rise to an appreciable direct exchange [40]. However, this situation can give rise to important superexchange interactions. These indirect exchange interactions depend on the bond length and bond angle. Moreover, indirect exchange interactions decrease with the magnetic ions separation and increase as the angle formed by triplet Fe$^{3+}$–O$^{2-}$–Fe$^{3+}$ tends to 180° [41]. Table 3 shows the bond angle between Fe1$^{3+}$–O$^{2-}$–Fe2$^{3+}$ decreases from 129.04° to 126.67° for Dy$^{3+}$, 127.67° for Nd$^{3+}$, 127° for Sm$^{3+}$, and 127.69° for Y$^{3+}$ doped Gd

garnet samples at $x = 0.75$. This arrangement favors weak superexchange interaction between the tetrahedral and octahedral sublattices. The bond length *Fe1-Fe2* decreased for $Dy^{3+}$ and $Y^{3+}$ and increased for $Nd^{3+}$ and $Sm^{3+}$ doped samples from 3.485 Å ($x = 0.0$) to 3.483 Å for $Dy^{3+}$ ($x = 0.75$), 3.478 Å for $Y^{3+}$ ($x = 0.75$), 3.494 Å for $Nd^{3+}$ ($x = 0.75$) and 3.492 Å for $Sm^{3+}$ ($x = 0.75$), respectively. Thus, the strength of superexchange interaction between $Fe^{3+}–O^{2-}–Fe^{3+}$ increased for $Dy^{3+}$ and $Y^{3+}$ doped and decreased for $Nd^{3+}$ and $Sm^{3+}$ doped garnet samples.

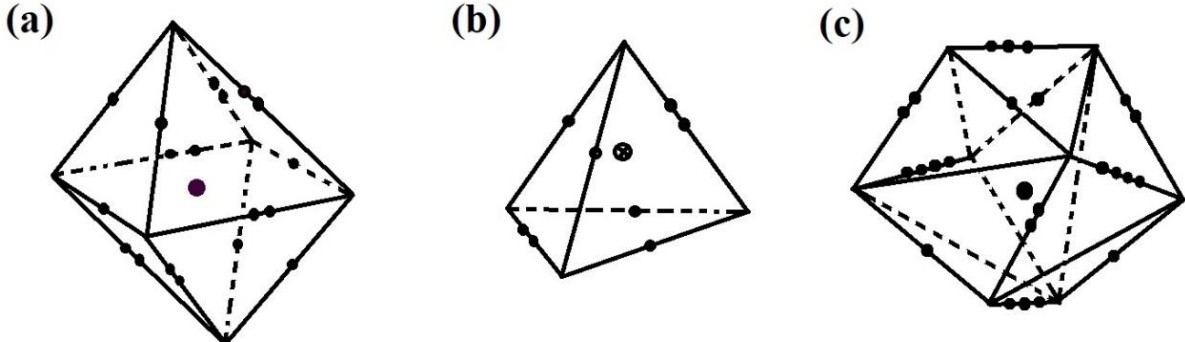

**Figure 8.** Position of the oxygen and magnetic ions at the (**a**) octahedral, (**b**) tetrahedral, and (**c**) dodecahedral sites.

The angle between $Fe1^{3+}–O^{2-}–Fe2^{3+}$ is greater than between $Gd^{3+}–O^{2-}–Fe1^{3+}$ and $Gd^{3+}–O^{2-}–Fe2^{3+}$. Therefore, the superexchange interaction $Fe1^{3+}–O^{2-}–Fe2^{3+}$ could be weaker than $Gd^{3+}–O^{2-}–Fe1^{3+}$ and $Gd^{3+}–O^{2-}–Fe2^{3+}$. The exchange interaction of two magnetic ions and $Gd^{3+}$ ions is through an oxygen ion bridging between them [42]. In this interaction, the overlap of the $2p$ electrons (with dumbbell-shaped distribution) of the oxygen ion with the electronic distribution of the magnetic ions is an important feature. The interaction increases with the overlap and, accordingly, will be greatest for short $Fe^{3+}/Gd^{3+}$-$O^{2-}$ distances and $Gd^{3+}/Fe1^{3+}$-$O^{2-}$-$Fe2^{3+}$ angles near 180°. Therefore, in $Gd_3Fe_5O_{12}$, the strongest interaction (Table 3A) will probably occur between $Fe^{3+}$(octa) and $Fe^{3+}$(tetra), for which the $Fe1^{3+}$-$O^{2-}$-$Fe2^{3+}$ angle is 129.04° and bond length is ~3.485 Å. In $Gd_{3-x}RE_xFe_5O_{12}$, the $RE^{3+}$-$O^{2-}$-$Fe^{3+}$(tetra) angle is 121.36° (Table 3A), and the bond length is 3.117 Å (Table 3B). Table 4 lists the atomic site occupancy for $Gd^{3+}$, $RE^{3+}$, $Fe^{3+}$, and $O^{2-}$ determined from the Rietveld refinement. As reported in Reference [43], the crystallographic structure is defined in the space group *Ia-3d* (#230) with four independent atoms: gadolinium, oxygen, and the two iron atoms successively at 24*c*, 96*h*, 16*a*, and 24*d* sites. Gd (dodecahedral site) has a maximum occupancy at $x = 0.0$, which decreases with the $x$ content of $RE^{3+}$. The compound's chemical formula derived from the atomic site occupancy is listed in Table 4. The derived composition of the compound matches the assumed starting stoichiometry of the compound.

**Table 3.** (**A**) Rietveld refinement and DFT derived bond-angle for $Gd_{3-x}RE_xFe_5O_{12}$ compound. (**B**) Rietveld refinement and DFT derived bond-distance for the $Gd_{3-x}RE_xFe_5O_{12}$ compound.

**(A)**

| $Gd_{3-x}RE_xFe_5O_{12}$ | | Bond Angle (°) | | | | | | | | | |
|---|---|---|---|---|---|---|---|---|---|---|---|
| | | Gd–O–Fe2(tetra) | | Gd–O–Fe2(tetra) | | Gd–O–Fe1(Octa) | | Gd–O–Fe1(Octa) | | Fe1–O–Fe2 | |
| | $x$ | Theory | Expt. | Theory | Expt. | Theory | Expt. | Theory | Expt. | Theory | Expt. |
| $Dy^{3+}$ | 0.00 | 93.37 | 92.36 | 122.95 | 121.36 | 101.54 | 101.78 | 104.17 | 104.43 | 126.19 | 129.04 |
| | 0.25 | 93.19 | 93.10 | 123.3 | 122.05 | 102.02 | 101.94 | 103.99 | 104.12 | 125.86 | 127.82 |
| | 0.50 | 93.18 | 91.87 | 123.11 | 122.51 | 101.52 | 102.46 | 103.99 | 103.66 | 125.92 | 127.66 |
| | 0.75 | 93.08 | 93.23 | 122.04 | 122.85 | 101.43 | 102.55 | 103.65 | 104.15 | 126.6 | 126.67 |
| $Nd^{3+}$ | 0.00 | 93.37 | 92.36 | 122.95 | 121.36 | 101.54 | 101.78 | 104.17 | 104.43 | 126.19 | 129.04 |
| | 0.25 | 93.38 | 93.36 | 122.72 | 121.19 | 101.71 | 100.82 | 103.95 | 104.73 | 127.26 | 128.36 |
| | 0.50 | 93.55 | 92.28 | 121.62 | 121.24 | 100.35 | 101.66 | 102.56 | 103.73 | 128.73 | 127.92 |
| | 0.75 | 93.41 | 91.21 | 122.03 | 121.29 | 100.88 | 101.66 | 102.67 | 104.88 | 128.2 | 127.03 |
| $Sm^{3+}$ | 0.00 | 93.37 | 92.36 | 122.95 | 121.36 | 101.54 | 101.78 | 104.17 | 104.43 | 126.19 | 129.04 |
| | 0.25 | 93.46 | 92.60 | 123.42 | 122.10 | 101.02 | 102.21 | 103.89 | 104.32 | 127.44 | 128.75 |
| | 0.50 | 93.21 | 91.71 | 122.11 | 122.32 | 100.84 | 102.48 | 103.36 | 104.04 | 128.13 | 128.34 |
| | 0.75 | 93.37 | 92.43 | 122.04 | 122.71 | 100.79 | 101.71 | 103.47 | 102.72 | 128.16 | 127.92 |
| $Y^{3+}$ | 0.00 | 93.37 | 92.36 | 122.95 | 121.36 | 101.54 | 101.78 | 104.17 | 104.43 | 126.19 | 129.04 |
| | 0.25 | 93.22 | 91.72 | 123.52 | 121.61 | 101.02 | 101.83 | 104.25 | 104.51 | 127.44 | 128.93 |
| | 0.50 | 93.53 | 92.95 | 123.56 | 121.71 | 100.86 | 101.25 | 103.83 | 104.33 | 127.64 | 128.82 |
| | 0.75 | 93.63 | 93.06 | 123.58 | 122.21 | 100.85 | 101.34 | 103.88 | 103.92 | 127.53 | 127.69 |

**(B)**

| $Gd_{3-x}RE_xFe_5O_{12}$ | | Bond Lengths (Å) | | | | | | | | | |
|---|---|---|---|---|---|---|---|---|---|---|---|
| | | Gd–Fe2(Tetra) | | Gd–Fe1(Octa) | | Fe2(Tetra)–O | | Fe1–Fe2 | | Fe1(Octa)–O | |
| | $x$ | Theory | Expt. | Theory | Expt. | Theory | Expt. | Theory | Expt. | Theory | Expt. |
| $Dy^{3+}$ | 0.00 | 3.117 | 3.117 | 3.485 | 3.485 | 1.878 | 1.876 | 3.484 | 3.485 | 2.029 | 1.984 |
| | 0.25 | 3.111 | 3.117 | 3.482 | 3.485 | 1.875 | 1.877 | 3.467 | 3.485 | 2.043 | 1.993 |
| | 0.50 | 3.104 | 3.116 | 3.456 | 3.483 | 1.867 | 1.884 | 3.462 | 3.4838 | 2.028 | 1.997 |
| | 0.75 | 3.102 | 3.115 | 3.456 | 3.483 | 1.864 | 1.875 | 3.454 | 3.483 | 2.015 | 2.022 |
| $Nd^{3+}$ | 0.00 | 3.117 | 3.117 | 3.485 | 3.485 | 1.878 | 1.876 | 3.484 | 3.485 | 2.029 | 1.984 |
| | 0.25 | 3.112 | 3.121 | 3.473 | 3.4888 | 1.876 | 1.872 | 3.469 | 3.4888 | 2.002 | 1.997 |
| | 0.50 | 3.121 | 3.123 | 3.466 | 3.4911 | 1.863 | 1.871 | 3.474 | 3.4911 | 1.996 | 1.998 |
| | 0.75 | 3.11 | 3.126 | 3.472 | 3.4944 | 1.863 | 1.889 | 3.464 | 3.4944 | 2.043 | 1.928 |
| $Sm^{3+}$ | 0.00 | 3.117 | 3.117 | 3.485 | 3.485 | 1.878 | 1.876 | 3.484 | 3.485 | 2.029 | 1.984 |
| | 0.25 | 3.113 | 3.119 | 3.462 | 3.4868 | 1.865 | 1.886 | 3.465 | 3.4868 | 1.997 | 2.007 |
| | 0.50 | 3.114 | 3.121 | 3.469 | 3.4896 | 1.865 | 1.891 | 3.471 | 3.4896 | 2.003 | 1.986 |
| | 0.75 | 3.115 | 3.123 | 3.472 | 3.4918 | 1.865 | 1.867 | 3.472 | 3.4918 | 1.995 | 2.034 |
| $Y^{3+}$ | 0.00 | 3.117 | 3.117 | 3.485 | 3.485 | 1.878 | 1.876 | 3.484 | 3.485 | 2.029 | 1.984 |
| | 0.25 | 3.111 | 3.116 | 3.467 | 3.4846 | 1.866 | 1.87 | 3.466 | 3.4846 | 1.999 | 1.981 |
| | 0.50 | 3.113 | 3.114 | 3.463 | 3.4813 | 1.865 | 1.864 | 3.475 | 3.4813 | 1.995 | 1.996 |
| | 0.75 | 3.108 | 3.111 | 3.463 | 3.4789 | 1.864 | 1.863 | 3.477 | 3.4789 | 1.998 | 2.012 |

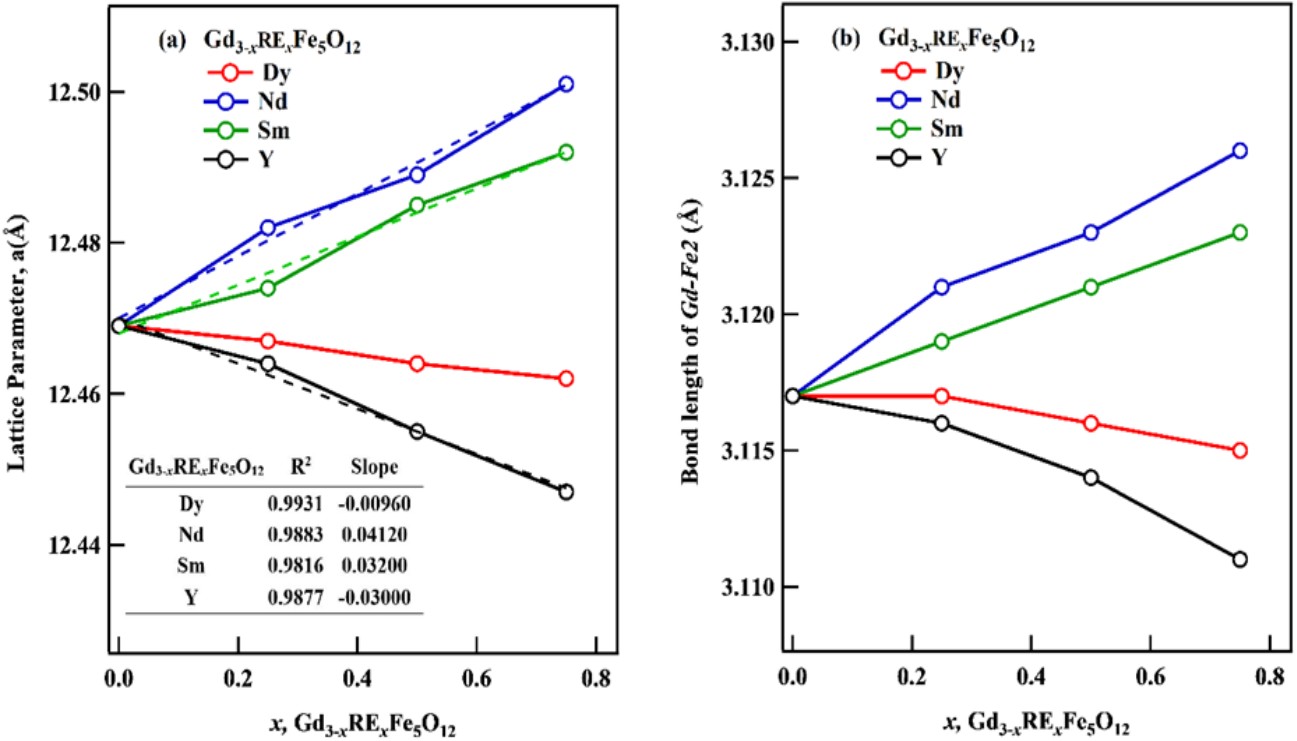

**Figure 9.** (**a**) Lattice parameter and (**b**) Gd–Fe2 bond length of the $Gd_{3-x}RE_xFe_5O_{12}$ compounds.

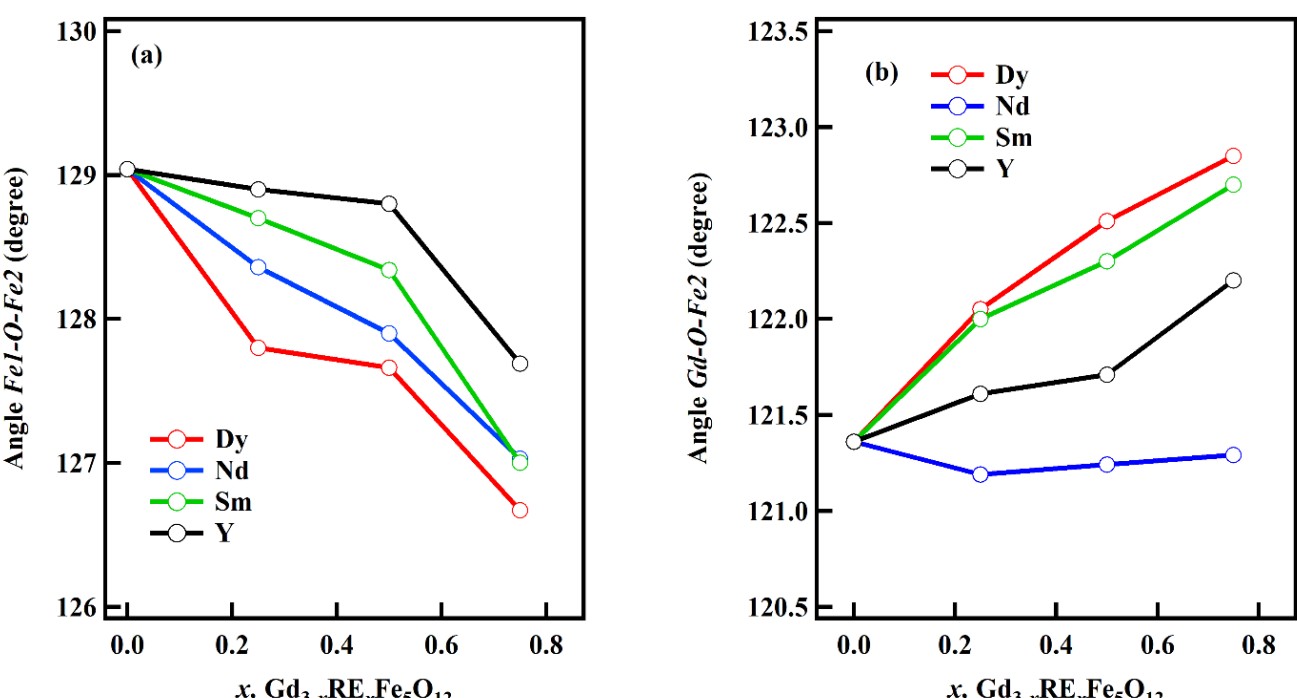

**Figure 10.** Bond angle (**a**) Fe1–O–Fe2 and (**b**) Gd–O–Fe2 for $Gd_{3-x}RE_xFe_5O_{12}$ compounds.

**Table 4.** Atomic site occupancy derived from Rietveld refinement for the $Gd_{3-x}RE_xFe_5O_{12}$ compound.

| $Gd_{3-x}RE_xFe_5O_{12}$ RE | $x$ | Gd | RE | Fe2(tetra.) | Fe1(Octa.) | Chemical Formula |
|---|---|---|---|---|---|---|
| $Dy^{3+}$ | 0.00 | 1.0052 | - | 0.9982 | 1.0093 | $Gd_{3.02}Fe_{2.99}Fe_{2.02}O_{12}$ |
| | 0.25 | 0.9264 | 0.0899 | 0.9923 | 1.0587 | $Gd_{2.78}Dy_{0.27}Fe_{2.98}Fe_{2.12}O_{12}$ |
| | 0.50 | 0.8288 | 0.1492 | 1.0006 | 1.0067 | $Gd_{2.49}Dy_{0.45}Fe_{3.00}Fe_{2.01}O_{12}$ |
| | 0.75 | 0.7784 | 0.2181 | 1.0033 | 1.0108 | $Gd_{2.34}Dy_{0.65}Fe_{3.01}Fe_{2.02}O_{12}$ |
| $Nd^{3+}$ | 0.00 | 1.0052 | - | 0.9982 | 1.0093 | $Gd_{3.02}Fe_{2.99}Fe_{2.02}O_{12}$ |
| | 0.25 | 0.9245 | 0.0806 | 0.9900 | 0.9917 | $Gd_{2.77}Nd_{0.24}Fe_{2.97}Fe_{1.98}O_{12}$ |
| | 0.50 | 0.849 | 0.1649 | 0.9985 | 1.0022 | $Gd_{2.55}Nd_{0.49}Fe_{3.00}Fe_{2.00}O_{12}$ |
| | 0.75 | 0.7438 | 0.2526 | 1.0022 | 1.0111 | $Gd_{2.23}Nd_{0.76}Fe_{3.01}Fe_{2.02}O_{12}$ |
| $Sm^{3+}$ | 0.00 | 1.0052 | - | 0.9982 | 1.0093 | $Gd_{3.02}Fe_{2.99}Fe_{2.02}O_{12}$ |
| | 0.25 | 0.9111 | 0.0829 | 1.011 | 1.0186 | $Gd_{2.73}Sm_{0.25}Fe_{3.03}Fe_{2.04}O_{12}$ |
| | 0.50 | 0.8215 | 0.1604 | 0.9946 | 1.0073 | $Gd_{2.47}Sm_{0.48}Fe_{2.98}Fe_{2.01}O_{12}$ |
| | 0.75 | 0.7688 | 0.2269 | 1.0576 | 1.049 | $Gd_{2.31}Sm_{0.68}Fe_{3.17}Fe_{2.10}O_{12}$ |
| $Y^{3+}$ | 0.00 | 1.0052 | - | 0.9982 | 1.0093 | $Gd_{3.02}Fe_{2.99}Fe_{2.02}O_{12}$ |
| | 0.25 | 0.9020 | 0.0873 | 1.0512 | 1.0705 | $Gd_{2.71}Y_{0.26}Fe_{3.15}Fe_{2.01}O_{12}$ |
| | 0.50 | 0.8171 | 0.1682 | 0.9917 | 0.9977 | $Gd_{2.45}Y_{0.51}Fe_{2.98}Fe_{2.00}O_{12}$ |
| | 0.75 | 0.7890 | 0.2156 | 1.0220 | 0.9900 | $Gd_{2.37}Y_{0.65}Fe_{3.07}Fe_{1.98}O_{12}$ |

### 3.2. Structural Parameters

The site radii ($r_A$ and $r_B$), bond length ($R_A$ and $R_B$), shared edges ($d_{AE}$ and $d_{BE}$) length, and unshared edges ($d_{BEU}$) length for tetrahedral and octahedral of $Gd_{3-x}RE_xFe_5O_{12}$ compound are calculated using Bertaut method [44]. The subscripts $A$ and $B$ refer to octahedral and tetrahedral sites.

$$r_A = \left[u - \frac{1}{4}\right]a\sqrt{3} - R_o \tag{1}$$

$$r_B = \left[\frac{5}{8} - u\right]a - R_o \tag{2}$$

$$R_A = a\sqrt{3}\left(Б + \frac{1}{8}\right) \tag{3}$$

$$R_B = a\sqrt{\left(\frac{1}{16} - \frac{Б}{2} + 3Б^2\right)} \tag{4}$$

$$d_{AE} = a\sqrt{2}\left(2u - \frac{1}{2}\right) \tag{5}$$

$$d_{BE} = a\sqrt{2}(1 - 2u) \tag{6}$$

$$d_{BEU} = a\sqrt{(4u^2 - 3u + 11/16)}, \tag{7}$$

where $R_o$ is the radius of the oxygen ion (1.32 Å), u is a positional parameter, $u_{ideal}$ is 0.375 Å, and $Б = u - u_{ideal}$, Б is the deviation of oxygen parameters [45]. The positional parameter ($u$) is calculated from the relation [46];

$$u = \frac{\frac{1}{2}R^2 - \frac{11}{12} + \left(\frac{11}{48}R^2 - \frac{1}{18}\right)^2}{2R^2 - 2}, \tag{8}$$

where $R = (Fe2\text{-}O)/(Fe1\text{-}O)$. The calculated $r_A$, $r_B$, $R_A$, $R_B$, $d_A$, $d_{BE}$, and $d_{BEU}$ values for RE$^{3+}$ doped garnets are listed in Table 5. The unshared edges, $d_{AE}$, and $d_{BE}$ for Dy$^{3+}$ and Y$^{3+}$ decreased, while for Nd$^{3+}$ and Sm$^{3+}$ doping, the value increased [47]. Similarly, site radii $r_A$ and bond-length $R_A$ decreased, and $r_B$ and $R_B$ increased for Dy$^{3+}$ and Y$^{3+}$ doping, while for Nd$^{3+}$ and Sm$^{3+}$, the opposite trend is observed. These variations in structural parameters are per ionic radii differences between doped RE$^{3+}$ and Gd$^{3+}$ ions. The $u$ value is observed to remain unaffected by doping due to the centrosymmetric structure of the compounds.

**Table 5.** Site-radii (r), bond-length (R), and share-edges (d) of the Gd$_{3-x}$RE$_x$Fe$_5$O$_{12}$ compound.

| RE | $x$ | $u$ | Site Radii | | Bond Length | | Shared Edges | | Unshared Edges | Tolerance Factor |
|---|---|---|---|---|---|---|---|---|---|---|
| | | | $r_A$ (Å) | $r_B$ (Å) | $R_A$ (Å) | $R_B$ (Å) | $d_{AE}$ (Å) | $d_{BE}$ (Å) | $d_{BEU}$ (Å) | $t$ |
| Dy$^{3+}$ | 0.00 | 0.382 | 1.542 | 1.704 | 2.862 | 3.026 | 4.673 | 4.143 | 4.412 | 0.669 |
| | 0.25 | 0.381 | 1.509 | 1.721 | 2.829 | 3.043 | 4.620 | 4.195 | 4.410 | 0.658 |
| | 0.50 | 0.382 | 1.536 | 1.705 | 2.856 | 3.028 | 4.664 | 4.149 | 4.410 | 0.668 |
| | 0.75 | 0.381 | 1.503 | 1.723 | 2.823 | 3.045 | 4.611 | 4.201 | 4.408 | 0.656 |
| Nd$^{3+}$ | 0.00 | 0.382 | 1.542 | 1.704 | 2.862 | 3.026 | 4.673 | 4.143 | 4.412 | 0.669 |
| | 0.25 | 0.382 | 1.528 | 1.716 | 2.848 | 3.038 | 4.651 | 4.175 | 4.416 | 0.663 |
| | 0.50 | 0.382 | 1.527 | 1.720 | 2.847 | 3.042 | 4.649 | 4.182 | 4.418 | 0.662 |
| | 0.75 | 0.385 | 1.617 | 1.672 | 2.937 | 2.998 | 4.796 | 4.043 | 4.427 | 0.694 |
| Sm$^{3+}$ | 0.00 | 0.382 | 1.542 | 1.704 | 2.862 | 3.026 | 4.673 | 4.143 | 4.412 | 0.669 |
| | 0.25 | 0.382 | 1.531 | 1.712 | 2.851 | 3.034 | 4.656 | 4.164 | 4.413 | 0.665 |
| | 0.50 | 0.383 | 1.559 | 1.700 | 2.878 | 3.023 | 4.701 | 4.127 | 4.418 | 0.674 |
| | 0.75 | 0.380 | 1.491 | 1.741 | 2.811 | 3.063 | 4.591 | 4.242 | 4.418 | 0.649 |
| Y$^{3+}$ | 0.00 | 0.382 | 1.542 | 1.704 | 2.862 | 3.026 | 4.673 | 4.143 | 4.412 | 0.669 |
| | 0.25 | 0.383 | 1.569 | 1.686 | 2.889 | 3.010 | 4.717 | 4.096 | 4.412 | 0.679 |
| | 0.50 | 0.382 | 1.515 | 1.714 | 2.835 | 3.036 | 4.629 | 4.177 | 4.406 | 0.661 |
| | 0.75 | 0.381 | 1.497 | 1.721 | 2.817 | 3.043 | 4.601 | 4.201 | 4.403 | 0.655 |

### 3.3. Crystallite Size and Density

The crystallite size and strain were also calculated using Halder-Wagner-Langford's (HWL) method [48]. The HWL equation relates the FWHM of peaks, $\beta$, with the mean crystallite size, "$D$," and the micro-deformation of a grain, $\varepsilon$ (strain parameter), as follows,

$$\left(\frac{\beta^*}{d^*}\right)^2 = \frac{1}{D}\left(\frac{\beta^*}{d^{*2}}\right) + \left(\frac{\varepsilon}{2}\right)^2, \tag{9}$$

where $\beta^*$ is given by $\beta^* = (\beta/\lambda)\cos(\theta)$, where $\lambda$ is the X-ray wavelength and $d^*$ is given as $d^* = (2/\lambda)\sin(\theta)$.

The plot of $(\beta^*/d^*)^2$ vs. $\beta^*/d^{*2}$ is a straight line, for which the intercept and the slope allow the values of the microstrain ($\varepsilon$) and the crystallite size ($D$) to be determined. The HWL plot has the advantage that data for reflections at low and intermediate angles are given more weight than those at higher diffraction angles, which are often less reliable. Figure 11 shows the HWL plots to compute the crystallite size and strain of the sample using Equation (2). The average crystallite size of the doped Gd$_{3-x}$RE$_x$Fe$_5$O$_{12}$ samples obtained from HWL plots is listed in Table 6. The crystallite size of the pure sample obtained from Scherrer's method was 67 nm and decreased with $x$ content from 67 nm to 64 nm for Dy$^{3+}$ ($x = 0.75$), 61 nm for Nd$^{3+}$ ($x = 0.75$), 63 nm for Sm$^{3+}$ ($x = 0.75$) and 53 nm for Y$^{3+}$ ($x = 0.75$) doped compound. The observed grain refinement upon RE$^{3+}$ substitution can result (1) from the increased microstrain due to the size difference between Gd$^{3+}$ and RE$^{3+}$ [49,50], (2) RE$^{3+}$ diffusion to the boundaries, which could restrain the grain growth [51], and (3) the reduction in the unit cell volume accompanied by shortening the diffusion path between nearby grains could result in smaller grains during calcination. Similar grain refinement is observed upon RE$^{3+}$ substitution in other ferrites [52,53]. The HWL strain increased with

$RE^{3+}$ content and reached a value of $2.77 \times 10^{-4}$, $1.68 \times 10^{-4}$, $7.11 \times 10^{-4}$, and $2.77 \times 10^{-4}$ for $x = 0.75$. The observed positive slopes in the HWL plots in all samples indicate the presence of strain. Due to the complex and inhomogeneous nature of the substituted oxide sample, the single origin of strain is difficult to pin. The observed strain may have its origin in ionic size differences, vacancies, and random distribution of ions on the atomic sites.

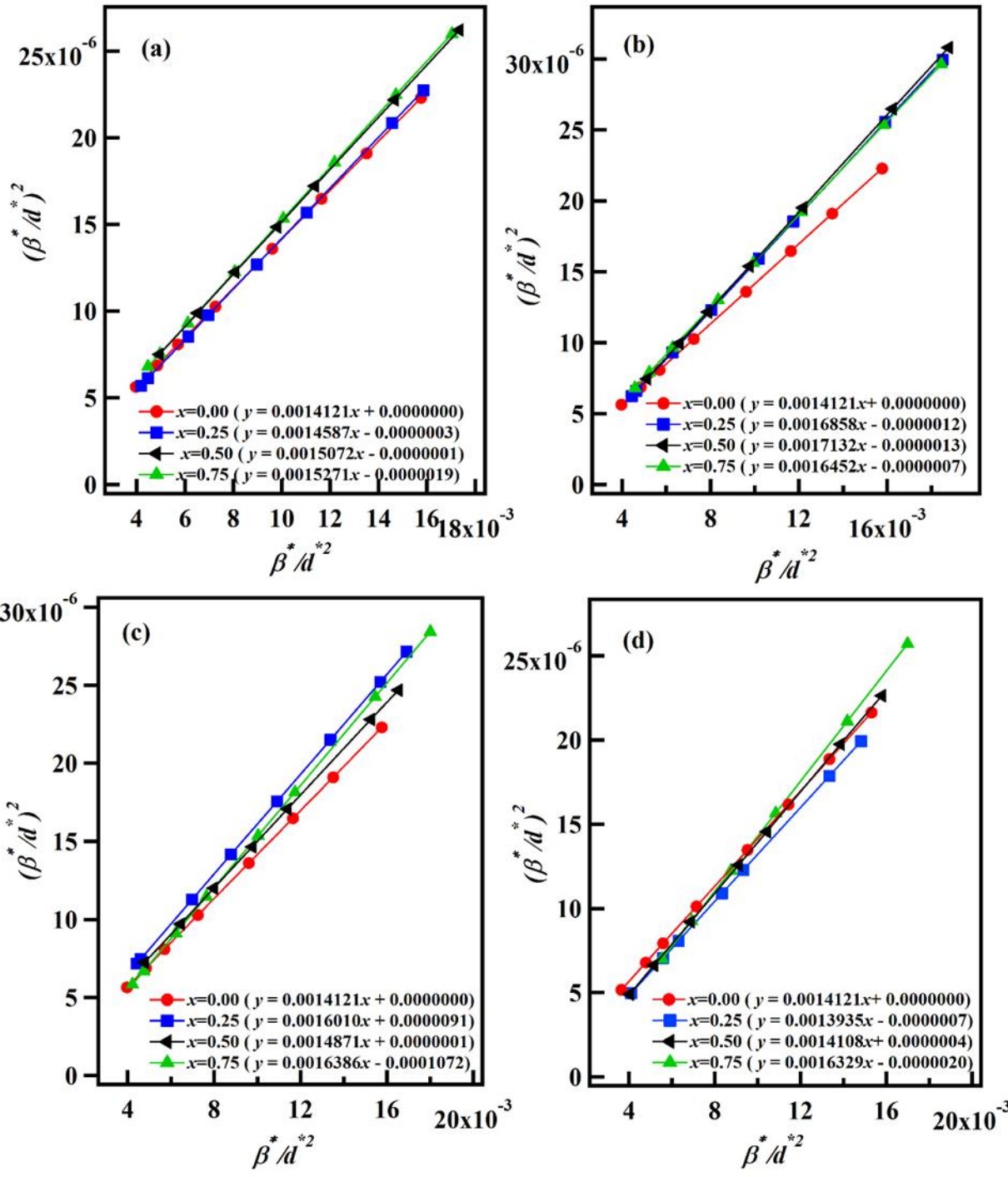

**Figure 11.** (**a**–**d**): HWL plots for the $Gd_{3-x}RE_xFe_5O_{12}$ compounds.

**Table 6.** Average crystallite size, strain, and X-ray density for the $Gd_{3-x}RE_xFe_5O_{12}$ compound.

| $Gd_{3-x}RE_xFe_5O_{12}$ | | Average Crystallite Size (nm) HWL Method | Grain Size from SEM (nm) | Strain from HWL Method $\times 10^{-4}$ | X-ray Density, $\rho$ (g/cm³) |
|---|---|---|---|---|---|
| **RE** | $x$ | | | | |
| $Dy^{3+}$ | 0.00 | 70.82 | 1000 | 3.7 | 6.46 |
| | 0.25 | 68.55 | | 12.5 | 6.47 |
| | 0.50 | 66.34 | | 5.6 | 6.48 |
| | 0.75 | 65.48 | 1000 | 2.8 | 6.49 |
| $Nd^{3+}$ | 0.00 | 70.82 | 1000 | 3.7 | 6.46 |
| | 0.25 | 59.32 | | 22.3 | 6.42 |
| | 0.50 | 58.68 | | 23.1 | 6.39 |
| | 0.75 | 60.78 | 500 | 16.8 | 6.34 |
| $Sm^{3+}$ | 0.00 | 70.82 | 1000 | 3.7 | 6.46 |
| | 0.25 | 62.46 | | 6.0 | 6.44 |
| | 0.50 | 67.24 | | 7.1 | 6.41 |
| | 0.75 | 61.03 | 1500 | 7.1 | 6.39 |
| $Y^{3+}$ | 0.00 | 70.82 | 1000 | 3.7 | 6.46 |
| | 0.25 | 70.81 | | 17.6 | 6.35 |
| | 0.50 | 70.88 | | 12.7 | 6.25 |
| | 0.75 | 74.70 | 1000 | 13.4 | 6.14 |

The X-ray density was calculated using the relation,

$$\rho_x = 8\,M/N_A a^3 \tag{10}$$

where $M$ is the relative molecular mass, $N_A$ is Avogadro's number, and $a$ is the lattice parameter. Table 6 listed the X-ray density for $Gd_{3-x}RE_xFe_5O_{12}$. The calculated density increased for the $Dy^{3+}$ (from 6.487 g/cm³ to 6.496 g/cm³), whereas it decreased for the $Nd^{3+}$, $Sm^{3+}$, and $Y^{3+}$ doped compounds. The change in density observed in the $RE^{3+}$ doped garnet is due to the doped element's different atomic radii and mass. The observed variation in X-ray density is due to the lower atomic mass of Nd (144.24 u), Sm (150.40 u), and Y (88.91 u), replacing Gd (157.20 u), and the higher atomic mass of Dy (162.50 u) replacing Gd in $Gd_{3-x}RE_xFe_5O_{12}$. However, the contribution of defects to the X-ray density cannot be ignored.

*3.4. Microstructural Analysis*

The surface morphologies of $Gd_{3-x}RE_xFe_5O_{12}$, $x = 0.0$, and 0.75 obtained via SEM are shown in Figure 12. The parent compound consists of irregularly shaped grains with well-defined boundaries and voids. The grain size measurement histogram is also shown in Figure 12. The average grain size, listed in Table 6, is obtained by fitting the size distribution histogram to the log-normal distribution function as reported by Odo [54].

$$f(d, \mu, \sigma) \;=\; \frac{1}{d\sigma\sqrt{2\pi}} exp\left[-\frac{(\ln(d) - \mu)^2}{2\sigma^2}\right], \tag{11}$$

where $d$ is the cross-sectional length of the particle, $\mu$ and $\sigma$ are the logarithmic mean and standard deviation, respectively. It is noted that the $Dy^{3+}$ and $Y^{3+}$ substitution garnet has a grain size similar to that of the parent compound $Gd_3Fe_5O_{12}$, whereas the grain size decreased upon $Nd^{3+}$ and increased with $Sm^{3+}$ substitution. The average length of particles for the $Gd_3Fe_5O_{12}$, $Dy^{3+}$ ($x = 0.75$), and $Y^{3+}$ ($x = 0.75$) substituted samples is ~1.0 µm, and the average length reduced to ~500 nm for the $Nd^{3+}$ ($x = 0.75$) doped samples. The observed decrease in particle size with $Nd^{3+}$ substitution can result from the increased microstrain due to the higher ionic radii of $Nd^{3+}$. Moreover, the diffusion of $Nd^{3+}$ to the boundaries

restrains grain growth. The average length of particles increased to 1.5 μm for the $Sm^{3+}$ ($x = 0.75$) samples. The increased grain size with $Sm^{3+}$ substitution is due to nearly the same ionic radii as $Gd^{3+}$, allowing the easy long-range diffusion of $Sm^{3+}$. However, the grain size also depends on the porosity, sintering temperature, and grain boundaries found in the substituted garnet compounds.

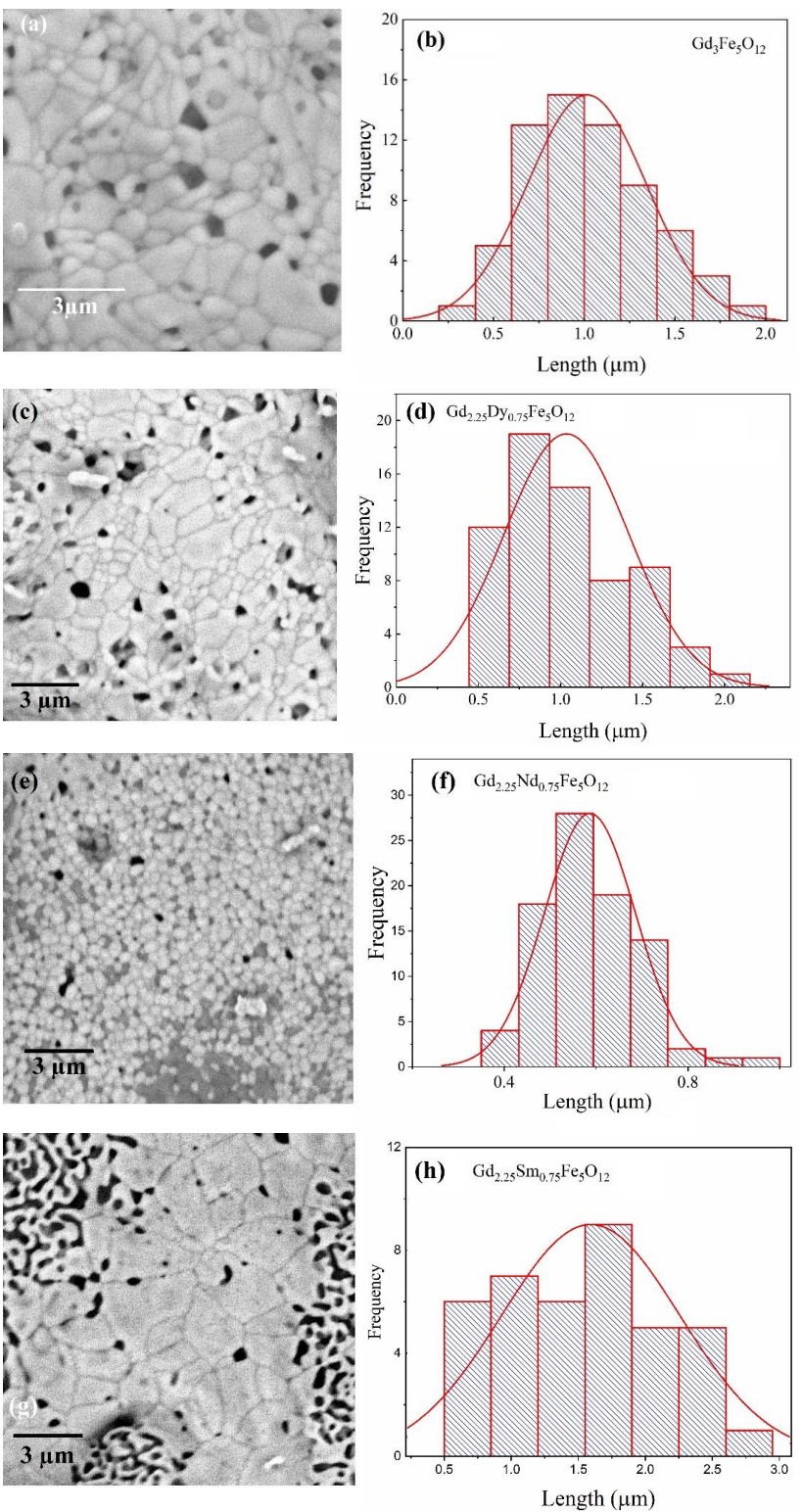

**Figure 12.** *Cont*.

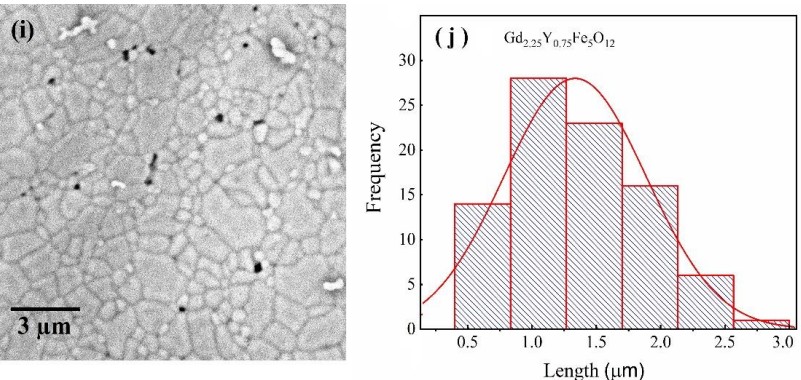

**Figure 12.** (**a–j**) SEM images and corresponding length distribution of the $Gd_{3-x}RE_xFe_5O_{12}$ compounds.

### 3.5. Theoretical Study

Ab initio density functional theory (DFT) calculations are performed using the VASP [33] simulation package for geometry optimization and post-processing calculations. The pseudopotential constructed under the projector augmented wave (PAW) [55] method describes the valence electrons. The exchange-correlation functional of the Perdew–Burke–Ernzerhof (PBE)+U [32,56] type is considered in the total energy calculations, where the wave function expansion is carried out by considering the plane-wave basis set having the energy cutoff of 400 eV. The spin and the orbital part of the total magnetic moment are calculated by considering the spin–orbit interactions. A gamma-centered k-point mesh sampled at $2 \times 2 \times 2$ is used to integrate the Brillouin zone. We used the total energy criteria for both electronic self-consistency and geometry optimization. The electronic self-consistency is achieved when the total energies of two consecutive electronic steps are smaller than $10^{-4}$ eV. The structures are allowed to relax along with the lattice parameters until the total energies of two consecutive ionic steps are smaller than $10^{-3}$ eV.

The total density of states (TDOS) of $Gd_3Fe_5O_{12}$ is shown in Figure 13a, along with the spin-up and spin-down components, and the orbital contributions from Gd, Fe, and O to the TDOS are shown in Figure 13b–d, respectively. From Figure 13b, it is clear that the *f*-orbital of the Gd atom has a major contribution to the spin-up component of the conduction band and the spin-down component of the valence band in TDOS, along with the small contribution from its *d*-orbital. Similarly, in Figure 13c,d, the *d*-orbitals of $Fe^{3+}$ atoms and the *p*-orbitals of the oxygen (O) seem to have a small contribution to the TDOS as well. The magnetism in the material arises because of the asymmetric nature of spin-up and spin-down components in the density of states, which is seen in Figure 14.

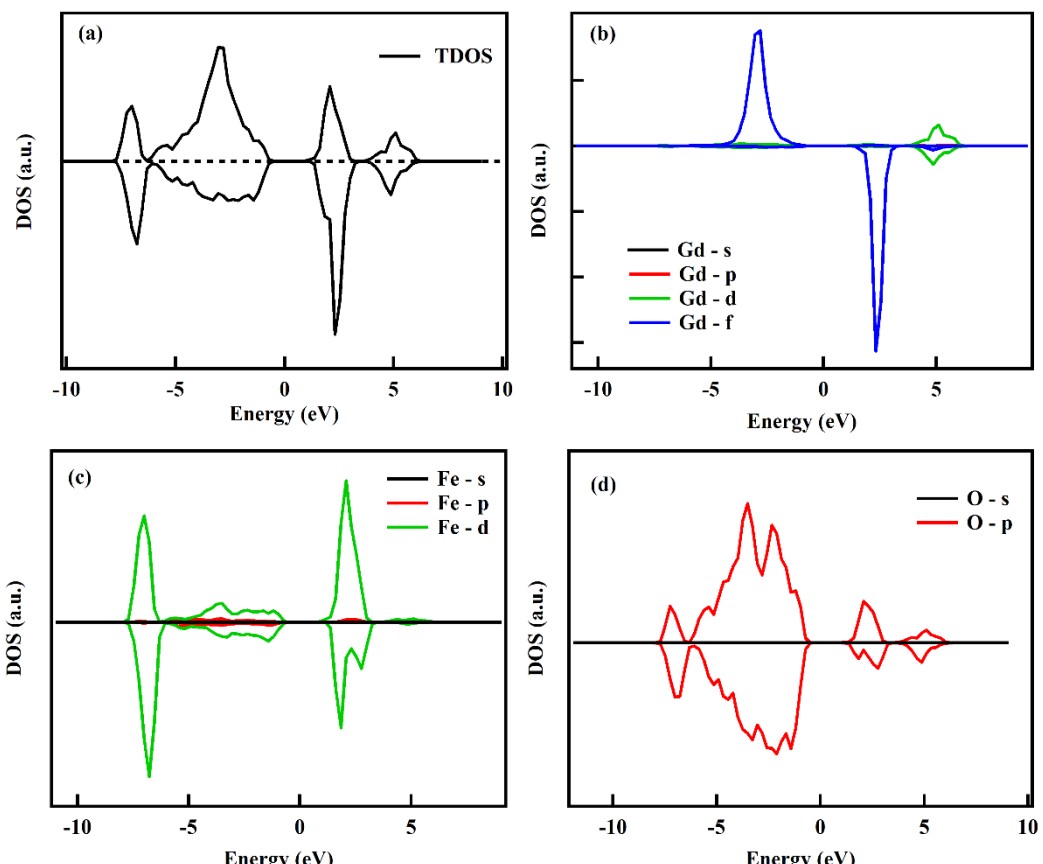

**Figure 13.** The partial and total density of states of pure $Gd_3Fe_5O_{12}$: (**a**) TDOS and (**b**–**d**) orbital contributions of individual elements to the total DOS. Both the spin-up and spin-down components are shown.

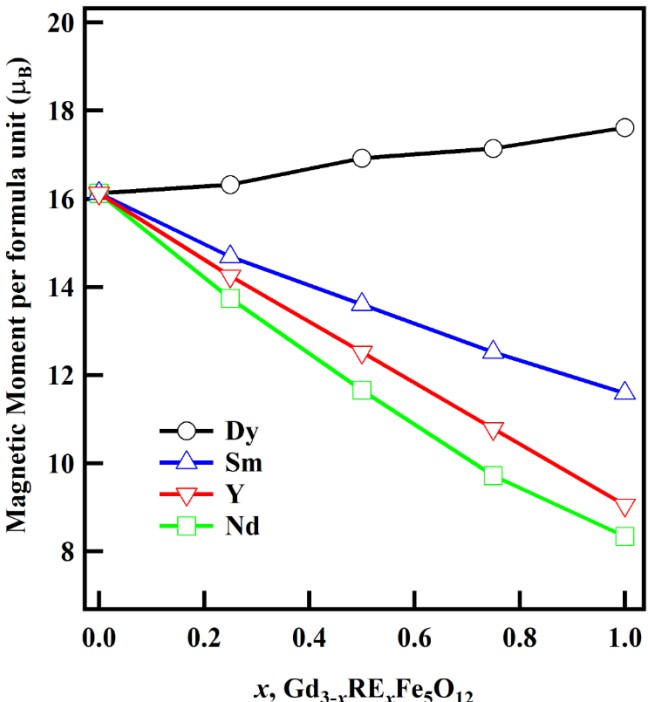

**Figure 14.** The magnetic moment of $Gd_{3-x}RE_xFe_5O_{12}$ as a function of doping content, *x*, was derived from the DFT study.

The $Gd_{3-x}RE_xFe_5O_{12}$ structure considered in the calculations consists of 80 atoms (Figure 1a), which is four times larger than its functional unit (f. u). The calculations are carried out for the four rare-earth (RE) elements, namely, Dy, Nd, Sm, and Y, which are used as a dopant on the Gd sites with values ranging from $x = 0$ to 1, with a step size of 0.25. The corresponding values for the total magnetic moments (spin and orbital) per formula unit of the optimized structures are obtained from the calculations. The effective Coulomb-exchange interaction ($U_{eff}$) value is set to be 6 eV for the $4f$ orbitals in Dy, Nd, Sm, and Gd, whereas it is 4 eV for the $d$ orbitals in Fe and Y [57,58]. The initial values of the magnetic moment of each element were taken from the literature [5,59,60]. The calculated values of the individual elements' orbital and spin magnetic moments after the optimization are listed in Table 7 below.

**Table 7.** DFT calculated values of orbital and spin magnetic moments of $RE^{3+}$, $Fe^{3+}$, and $O^{2-}$ ions.

| Ions | Orbital Moment ($\mu_L$) | Spin Moment ($\mu_S$) | Total Moment ($\mu_T$) |
|---|---|---|---|
| $Dy^{3+}$ | 4.07 | 4.98 | 9.05 |
| $Nd^{3+}$ | −4.43 | 2.89 | −1.54 |
| $Sm^{3+}$ | −2.43 | 4.98 | 2.55 |
| $Y^{3+}$ | 0.00 | 0.00 | 0.00 |
| $Gd^{3+}$ | 0.00 | 6.99 | 6.99 |
| $Fe^{3+}$ (Tetra) | 0.00 | −4.06 | −4.06 |
| $Fe^{3+}$ (Octa) | 0.00 | 4.16 | 4.16 |
| $O^{2-}$ | 0.00 | 0.00 | 0.00 |

From Table 7, the orbital magnetic moment of $Dy^{3+}$ has a positive value, whereas $Nd^{3+}$ and $Sm^{3+}$ have negative values. The higher negative $\mu_L$ value in $Nd^{3+}$ makes the total moment ($\mu_T$) negative. Moreover, $Fe^{3+}$, devoid of the orbital moment, has only the spin magnetic moment ($\mu_S$), with octahedral $Fe^{3+}$ having a positive value and tetrahedral $Fe^{3+}$ having a negative value. The elements $Y^{3+}$ and $O^{2-}$ have zero magnetic moments. At low temperatures with a $3d^5$, S = 5/2 configuration, we expect the magnetic moment to be 5 μB per iron ion. The Fe sublattices are anti-ferromagnetically coupled, giving a maximum net magnetization of 3 × (−5 μB) = −15 μB for the tetrahedral sublattice and 2 × (5 μB) = 10 μB for the octahedral sublattice per formula unit. The spontaneous magnetization direction of the Gd sublattice is taken to be positive. From the Gd sublattice, with $Gd^{3+}$, S = 7/2 ions, we can expect a maximum magnetization of 3 × 7μB = 21 μB. This gives the expected saturation magnetization of 16 μB for Gd garnet, a value confirmed experimentally earlier [28].

The variation of the magnetic moment formula unit as a function of doping concentration, $x$, is shown in Figure 14 for four different $RE^{3+}$ ions mentioned above. When $x = 0.0$, i.e., in the $Gd_3Fe_5O_{12}$ sample, the $\mu_B \sim 16$ agrees with the previous calculations [41,61,62]. With the increase in the $x$ value, the total magnetic moment increases in the case of $Dy^{3+}$ doping. This is because the magnetic moment of $Dy^{3+}$ is larger than that of $Gd^{3+}$. Meanwhile, the magnetic moment decreases in the other three doping cases (Nd, Sm, and Y). The decreasing rate is consistent with the order of their magnetic moments (Nd < Y < Sm), which is also consistent with the experimental results. Furthermore, structural parameters for $Gd_{3-x}RE_xFe_5O_{12}$, Table 3A, B, derived from DFT calculations, validate the values obtained from the XRD Rietveld refinement

### 3.6. Magnetic Properties

The Curie temperature *Tc* for $Gd_{3-x}RE_xFe_5O_{12}$ compounds was measured using a thermogravimetric analyzer (TGA) with a permanent magnet (Figure 15a–d). Figure 15 shows that the weight of the sample increased with temperature due to increased magnetic force on the sample. This implies that the net magnetic moment of the sample increased with the temperature up to *Tc*. The tetrahedral site iron ions have a positive moment, while

the octahedral site has a negative moment. At elevated temperatures due to thermal energy, these moments are canted with respect to the z-axis. The sum of the projection of these moments on the z-axis determines the net moment of the compound. The observed increase in magnetization with temperature indicates that the octahedral site moment dominates the net moment. At temperature $Tc$, thermal energy dominates the magnetic spins, and the material exhibits paramagnetic behavior. The observed $T_C$ value is 570 K for $x = 0.0$ and remains unaltered upon $RE^{3+}$ doping. The $Tc$ value is dictated by the strength and the number of superexchange $Fe^{3+}$–$O^{2-}$–$Fe^{3+}$ interactions, which largely remain unaffected by the $RE^{3+}$ doping. The observed $Tc$ with $RE^{3+}$ substitution is listed in Table 8.

The magnetization as a function of temperature for $RE^{3+}$ doped $Gd_{3-x}RE_xFe_5O_{12}$ was investigated in the temperature range below room temperature. Figure 16 displays the temperature dependence of magnetization $M(T)$ curves for $Gd_{3-x}RE_xFe_5O_{12}$ under zero-field cooled (ZFC) and field-cooled (FC) conditions at a 100 Oe field. A distinct characteristic of $M(T)$ curves is the presence of a compensation temperature, $T_{comp}$. The temperature at which the magnetization crosses zero is called the compensation temperature, $T_{comp}$. This occurs because the magnetization of $RE^{3+}$ ion at the $c$ sites is equal and opposite to the net magnetization of $Fe^{3+}$ ion sublattice at the $a$ and $d$ sites. i.e., $3Fe(d)-[2M_{Fe}(a) + 3M_{RE}(c)]$. Table 8 lists the comparison of $T_{comp}$ values for different doped garnet compounds. In the case of $Gd_{3-x}RE_xFe_5O_{12}$, $T_{comp}$ values vary between 280 K and 238 K, depending upon the type of $RE^{3+}$ doping.

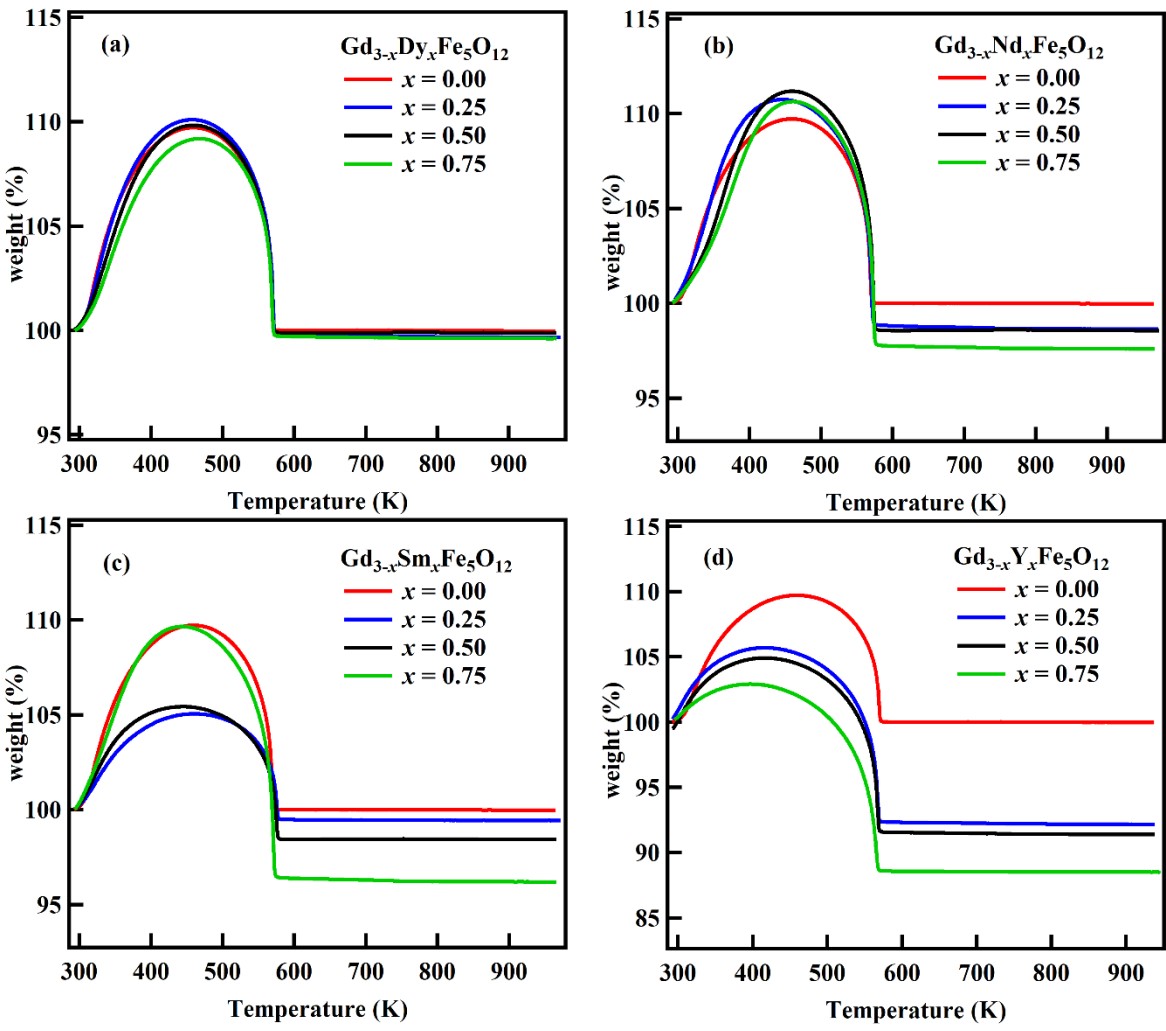

**Figure 15.** (a–d) Thermogravimetric curves of $Gd_{3-x}RE_xFe_5O_{12}$ compounds.

**Table 8.** $T_{comp}$ and $T_c$ values of garnet compounds.

| Compounds | $T_{comp}$ (K) | $T_C$ (K) | Reference |
|---|---|---|---|
| $Er_3Fe_5O_{12}$ | 87 | | [63] |
| $Er_3Fe_{4.2}Al_{0.8}O_{12}$ | 139 | | [63] |
| $Tb_3Fe_5O_{12}$ | 244 | | [64] |
| $Ho_3Fe_5O_{12}$ | 137 | | [65] |
| $Gd_3Fe_5O_{12}$ | 288 | | [66] |
| $Gd_3Fe_5O_{12}$ | 280 | 570 | Present work |
| $Gd_{2.25}Dy_{0.75}Fe_5O_{12}$ | 287 | 572 | Present work |
| $Gd_{2.25}Nd_{0.75}Fe_5O_{12}$ | 263 | 574 | Present work |
| $Gd_{2.25}Sm_{0.75}Fe_5O_{12}$ | 260 | 572 | Present work |
| $Gd_{2.25}Y_{0.75}Fe_5O_{12}$ | 238 | 567 | Present work |

This behavior may be explained in terms of the temperature dependence of the magnetization of the three magnetic sublattices (dodecahedral, octahedral, and tetrahedral) balance each other [61,62]. Neel et al. [41] reported that the magnetic properties of $RE_3Fe_5O_{12}$ can be explained by assuming that the three sublattice ferrimagnetism is due to positive $RE^{3+}$ spin on dodecahedral sites, positive $Fe^{3+}$ spin on octahedral site, and negative $Fe^{3+}$ ion tetrahedral sites. The exchange interaction between $RE^{3+}$ ions is almost negligible, so the moment of $RE^{3+}$ ions should be aligned with the exchange interaction with $Fe^{3+}$ ions. Structural evidence favors the strong magnetic interactions of the rare-earth ions with the tetrahedral $Fe^{3+}$ ions (down magnetic moment) [41]. The strong interaction of the $Gd^{3+}$ ion moments with those of the tetrahedral $Fe^{3+}$ ion moments leads to random canting of the $Gd^{3+}$ ion moments, thereby reducing the contribution of the dodecahedral sublattice to the net spontaneous magnetization of the garnet [67]. This could be the reason for the slow increase in the net magnetization value with temperature lowering.

A cusp is observed in the ZFC curves at a temperature where the $RE^{3+}$ moment aligns parallel to the net $Fe^{3+}$ moment. For $Gd_3Fe_5O_{12}$, the cusp is observed at temperature $(T_F)$ = 50 K, while for other $RE^{3+}$ substituted compounds, $T_F$ shifts to higher temperatures. Further, a decrease in magnetization value is observed at a temperature below $T_F$. This decrease in magnetization value occurs because of a strong $RE^{3+}$-$O^{2-}$-$Fe^{3+}$(Tetra.) interaction, which flips $RE^{3+}$ moments parallel to the $Fe^{3+}$(Tetra.) in a negative direction. The exchange interaction between the $4f$ rare-earth electrons and the $3d$ iron electrons is not direct but occurs indirectly via the oxygen ions [68]. This interaction is strengthened in the presence of $RE^{3+}$ ions with non-zero orbital angular momenta, such as $Nd^{3+}$, $Sm^{3+}$, and $Dy^{3+}$. This conclusion is further corroborated by noticing the absence of a cusp in $M$ vs. $T$ for $Y^{3+}$ doped garnet, where $Y^{3+}$ does not possess any orbital angular momentum. The increase in $T_F$ value with $RE^{3+}$ substitution results from the increased number and strength of superexchange interactions $RE^{3+}$–$O^{2-}$–$Fe^{3+}$(Tetra.) ensuing from increased bond-angle in favor of strengthening the interaction, Figure 10b. The $M$ vs. $T$ curve cusp is more prominent for the $Dy^{3+}$ doped sample. Because of negative moments of $Nd^{3+}$ ($-1.54$ µB), the magnetization attains a negative value below $T_{comp}$. Meanwhile, Sm3+ (2.55 µB) shows a positive magnetization value with a cusp below $T_{comp}$. This discussion concludes that $RE^{3+}$ with a finite orbital angular momentum couple strongly with $Fe^{3+}$ sublattice moment via superexchange interaction.

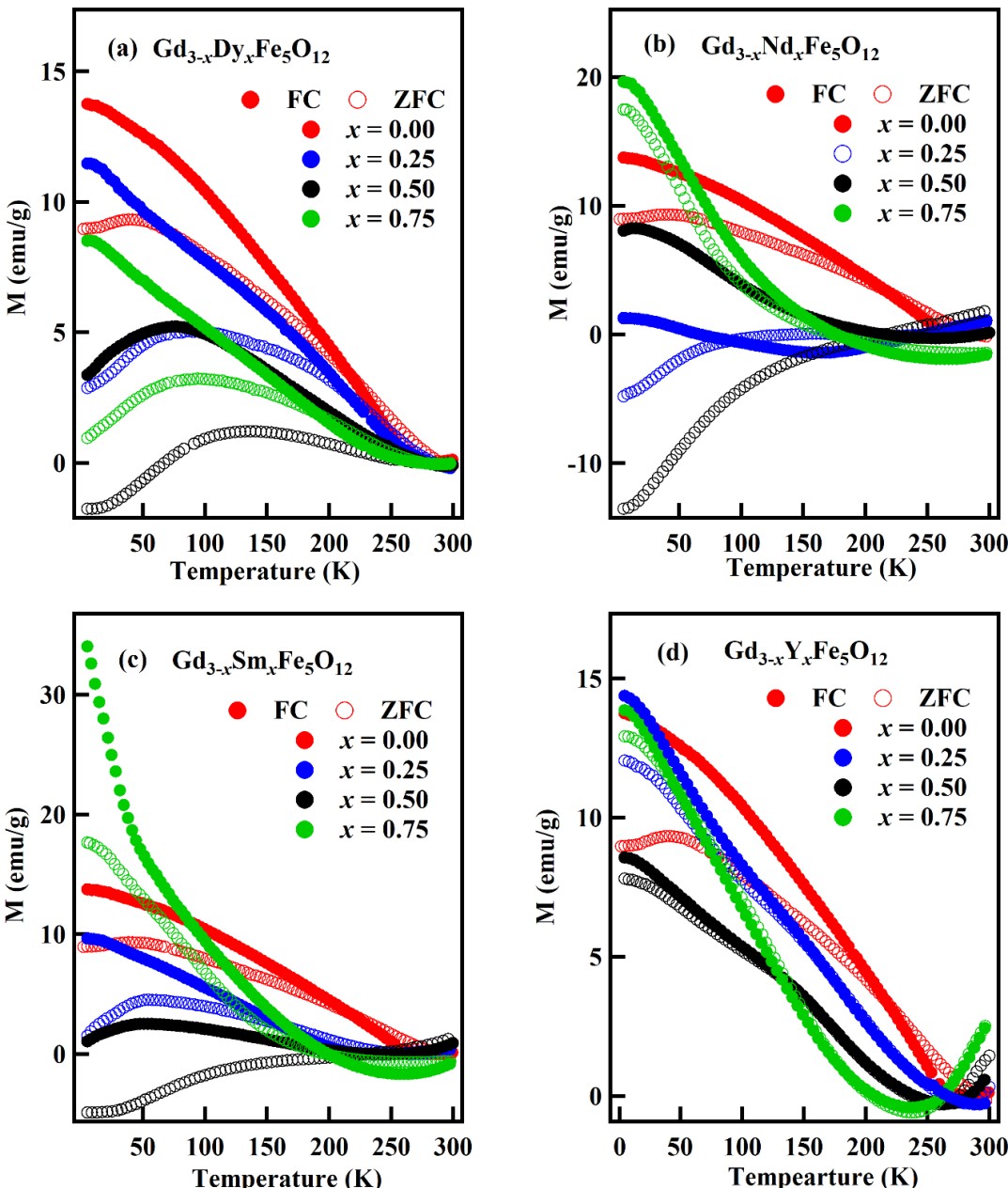

**Figure 16.** (**a–d**) FC/ZFC magnetization vs. temperature curves for the $Gd_{3-x}RE_xFe_5O_{12}$ compound.

The temperature-dependent magnetization process is depicted in Figure 17. At $T_{comp}$, the three sublattice moment adds to zero moments. Below $T_{comp}$, magnetization slowly peaks with rare-earth contributing positively to the net moment, while at low temperatures below $T_F$, increased $RE^{3+}$ moments canting due to strong $RE^{3+}$–$O^{2-}$–$Fe^{3+}$(Tetra.) superexchange interaction leads to net negative moments. The $T_F$ value shifts to a higher temperature depending on the strength and number of superexchange interaction pairs. At $x = 0.75$, at low temperatures, below $T_{comp}$, the magnetic anisotropy of $Nd^{3+}$, $Sm^{3+}$, and $Gd^{3+}$ exceeds that of iron, with their moment being aligned along the easy axis, thus increasing the net moment of the compound.

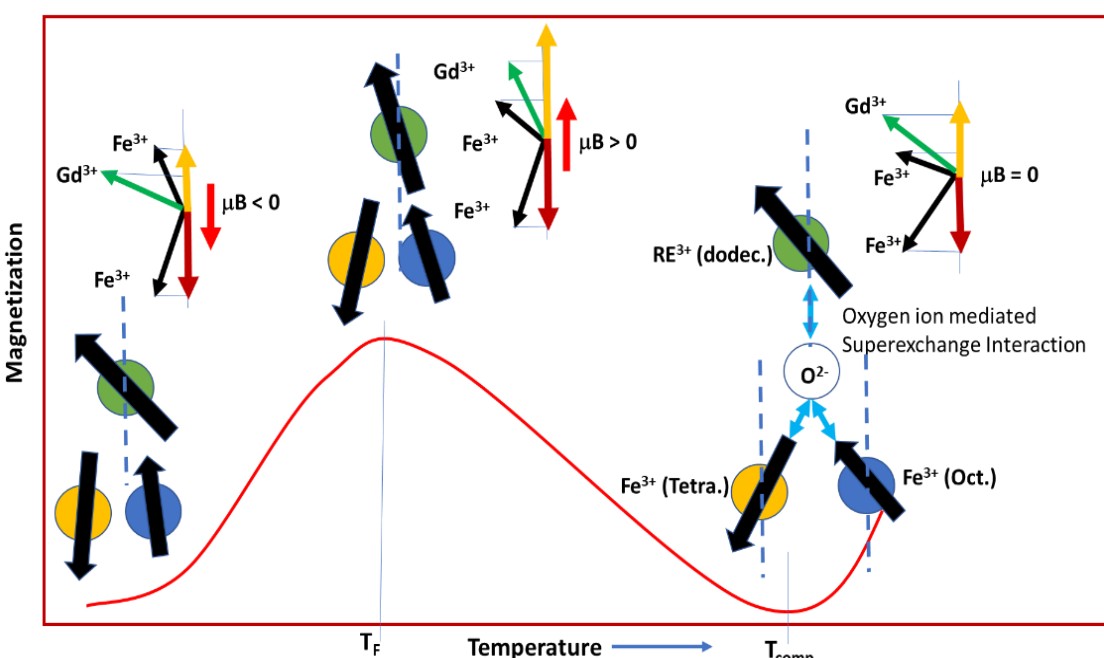

**Figure 17.** Schematic of three sublattice magnetizations as a function of temperature for $RE_3Fe_5O_{12}$ garnet compound.

Figure 18a–d shows the magnetization vs. field curves, *M* vs. *H*, for $Gd_{3-x}RE_xFe_5O_{12}$ measured at 5 K. All samples show the ferromagnetic behavior at 5 K. The magnetic curve tends to saturate below the applied field of 0.5 T for all samples. The saturation magnetization, *Ms*, value reached ~92.3 emu/g for *x* = 0.0 and decreased with *x* content for $Nd^{3+}$, $Sm^{3+}$, and $Y^{3+}$ except for $Dy^{3+}$. The saturation magnetization value for all samples matched the trend of theoretically derived values in Table 9. The total magnetic moment in the $Gd_{3-x}RE_xFe_5O_{12}$ garnet is due to the contribution of three different magnetic sublattices. The total magnetic moment is represented as:

$$3M_{Fe}(\text{tetra}) - [2M_{Fe}(\text{octa}) + (3-x)M_{Gd}(\text{dodec}) + xM_{RE}(\text{dodec})] \tag{12}$$

**Table 9.** Magnetic properties of $Gd_{3-x}RE_xFe_5O_{12}$. *K1* and *a2* are calculated using Equations (14) and (15).

| $Gd_{3-x}RE_xFe_5O_{12}$ | | *Ms* (emu/g) | Bohr Magneton | (α)Y-K (Degrees) |
|---|---|---|---|---|
| | *x* | | (μB) | |
| | 0.00 | 92.3 | 2.40 | - |
| | 0.25 | 90.1 | 2.35 | 12.70 |
| $Dy^{3+}$ | 0.50 | 94.8 | 2.47 | 40.08 |
| | 0.75 | 99.3 | 2.59 | 55.34 |
| | 0.25 | 79.4 | 2.08 | 21.11 |
| $Nd^{3+}$ | 0.50 | 70.6 | 1.84 | 48.13 |
| | 0.75 | 61.7 | 1.62 | 64.83 |
| | 0.25 | 88.0 | 2.29 | 14.89 |
| $Sm^{3+}$ | 0.50 | 76.8 | 2.01 | 46.07 |
| | 0.75 | 73.6 | 1.91 | 62.15 |
| | 0.25 | 85.6 | 2.22 | 17.23 |
| $Y^{3+}$ | 0.50 | 81.0 | 2.11 | 44.81 |
| | 0.75 | 70.5 | 1.84 | 62.74 |

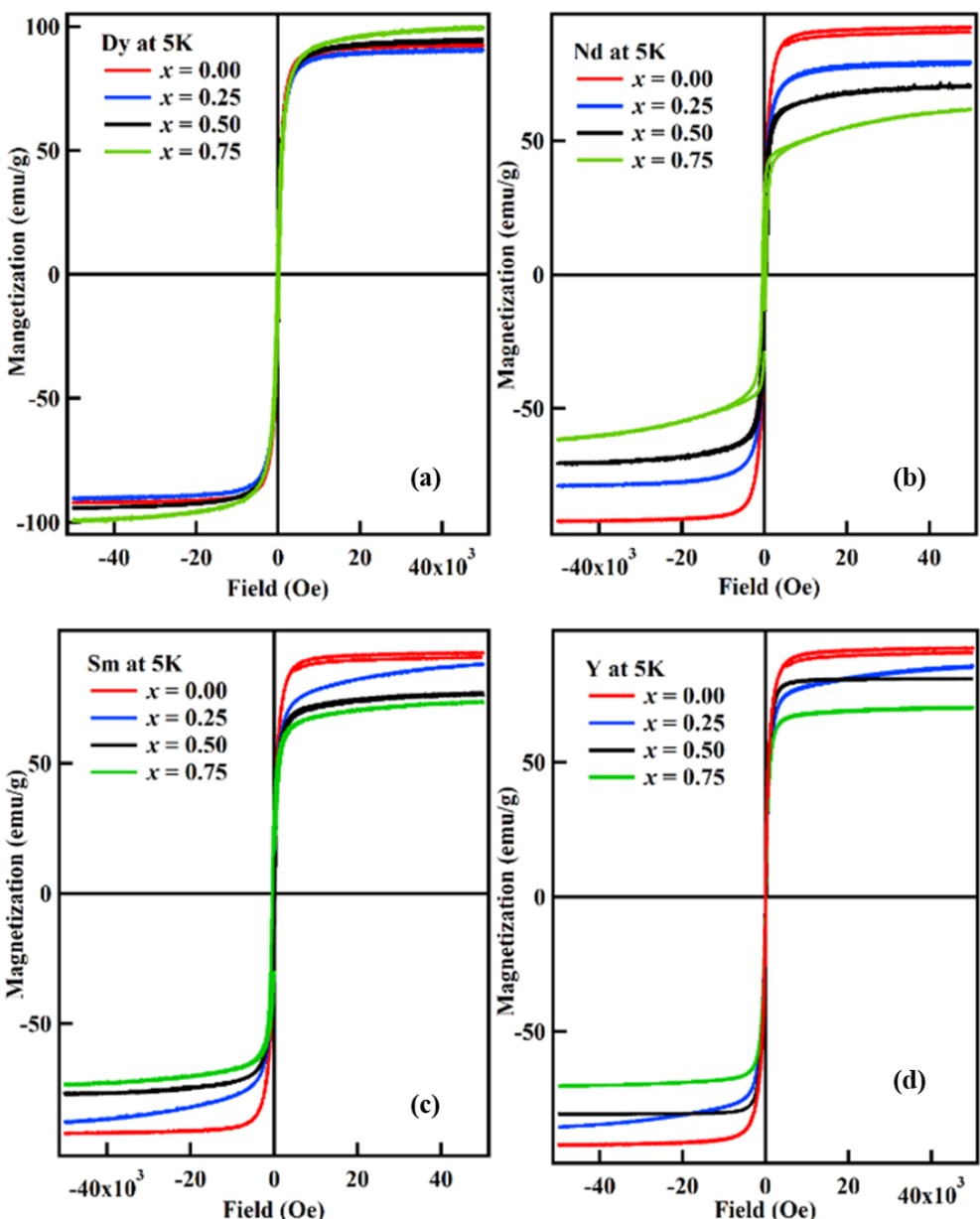

**Figure 18.** (a–d) Magnetization vs. field ($M$ vs. $H$) curves for $RE^{3+}$ doped $Gd_{3-x}RE_xFe_5O_{12}$ compounds at 5 K.

As obtained from the DFT study, the magnetic moment of the $Dy^{3+}$ ion has a maximum value (10.48 $\mu_B$), and low values for $Nd^{3+}$ (3.62 $\mu_B$), $Sm^{3+}$ (0.65 $\mu_B$), and $Y^{3+}$ (0 $\mu_B$) compared to $Gd^{3+}$ (6.9 $\mu_B$). Therefore, the net moment for the $Gd_{3-x}RE_xFe_5O_{12}$ compound decreases for $Nd^{3+}$, $Sm^{3+}$, and $Y^{3+}$ doped garnet but improves with $Dy^{3+}$ doping. The experimental magnetic moment of the $RE^{3+}$ substitution $Gd_{3-x}RE_xFe_5O_{12}$ sample is calculated in terms of Bohr magneton using the equation below and listed in Table 9.

$$\text{Bohr magneton } (\mu_B) \ = \ \frac{M \times M_s}{5585 \times \rho_{x-ray}} \tag{13}$$

where $\rho_{x-ray}$ is the X-ray density Equation (11), where $M$ is the molecular weight of the samples, and $M_s$ is the saturation magnetization in emu/g of $Gd_{3-x}RE_xFe_5O_{12}$. Bohr magneton value for $Gd_3Fe_5O_{12}$ is observed at 2.40 µB and changes with $RE^{3+}$ substitution. The magnetic moment of $Dy^{3+}$ doped garnet increases from 2.4 µB to 2.59 µB, whereas $Nd^{3+}$, $Sm^{3+}$, and $Y^{3+}$ doped samples show a decreasing trend. Change in Bohr magneton

value with $RE^{3+}$ substitution is due to the different magnetic moment of $RE^{3+}$ ions and matches the theoretical study's trend.

Yafet and Kittle (Y-K) angles describe the direction of spin of iron ions in ferrites. The Yafet and Kittle ($\alpha_{Y-K}$) angles of $RE^{3+}$ doped $Gd_{3-x}RE_xFe_5O_{12}$ are calculated by using the following equation:

$$\mu_B = (6+x)Cos\alpha_{Y-K} - 5(1-x), \tag{14}$$

where $\mu_B$ is the Bohr magnetons calculated experimentally. The $\alpha_{Y-K}$ arises due to the non-collinear direction of the moment between tetrahedral and octahedral sublattices. Table 9 shows the linear increase in the $\alpha_{Y-K}$ angle with $RE^{3+}$ substitution. In addition, $RE^{3+}$ doped samples show Y-K type canting of local moments. The linear trend in the $\alpha_{Y-K}$ angle with Re$x$ is due to the split of sublattices having magnetic moments equal in magnitude and each making an angle $\alpha_{Y-K}$ with the direction of net magnetization.

Figure 19a–d shows the *M* vs. *H* curves for $Gd_{3-x}RE_xFe_5O_{12}$ powder at 300 K. With the increase in temperature to 300 K, due to thermal fluctuation, the ferrimagnetic order is lost, and the $Gd_{3-x}RE_xFe_5O_{12}$ system attains paramagnetic order (it is not PM, it is unusual that there is remanence magnetization). To further investigate the effect of $RE^{3+}$ on the magnetic and magnetocaloric behaviors of $Gd_{3-x}RE_xFe_5O_{12}$, isothermal magnetization as a function of the applied field, M(H), was measured from 11 K to 210 K with a temperature step of 7 K up to 3T field. The isothermal plots for $Dy^{3+}$, $Nd^{3+}$, $Sm^{3+,}$ and $Y^{3+}$ substitution $Gd_{3-x}RE_xFe_5O_{12}$ are shown in Figures 20–23. The isothermal magnetization curve shows the ferromagnetic ordering at low temperatures and paramagnetic at elevated temperatures. The magnetization increases sharply and saturates immediately at the low field, which is a sign of ferromagnetic behavior. Magnetization increases gradually with an increasing field and does not show any sign of saturation, thus displaying the paramagnetic behavior.

### 3.7. Magnetocaloric Study

Our primary focus is to study the magnetocaloric effect of $RE^{3+}$ doped $Gd_{3-x}RE_xFe_5O_{12}$. The change in magnetic entropy ($\Delta S_M$) is the most recommended parameter to evaluate the efficiency of magnetocaloric materials. It is calculated using the magnetic isothermal data (Figures 20–23) near the vicinity of the transition temperature. The isothermal magnetic entropy change has been computed using the thermodynamic Maxwell relation [69],

$$\Delta S_M = \mu_o \int_{H_i}^{H_f} \left(\frac{\partial M}{\partial T}\right)_H dH \tag{15}$$

$$\Delta S_M = \frac{\mu_o}{\Delta T}\left[\int_0^{H_f} M(T+\Delta T, H)dH - \int_0^{H_f} M(T, H)dH\right] \tag{16}$$

It is numerically calculated as;

$$-\Delta S_M(H, T) = \sum \frac{M_i - M_{i+1}}{T_{i+1} - T_i}\Delta H_i, \tag{17}$$

where $H_i$ and $H_f$ are the initial and final external applied fields, and $\mu_o$ is the permeability of free space. $-\Delta S_M$ is calculated from the isothermal magnetization curve of Figures 20–23.

The magnetic entropy change has a maximum value near transition temperature, $T_c$, and its value decreases with a further increase or decrease in temperature. The sign of the magnetic entropy change is negative, which means heat is released when the magnetic field is changed adiabatically [70].

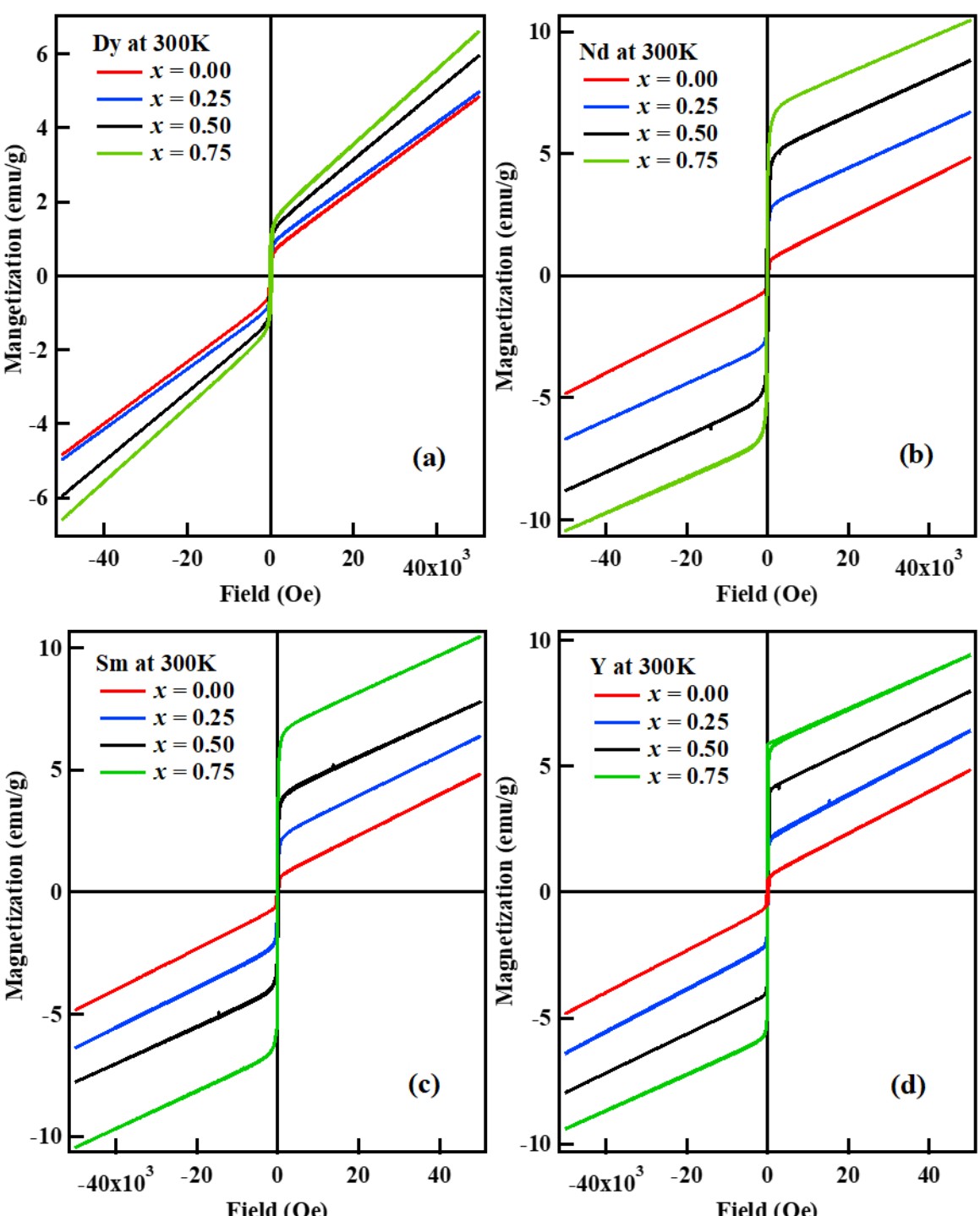

**Figure 19.** (**a**–**d**) M vs. H plots for the $Gd_{3-x}RE_xO_{12}$ compound measured at 300 K.

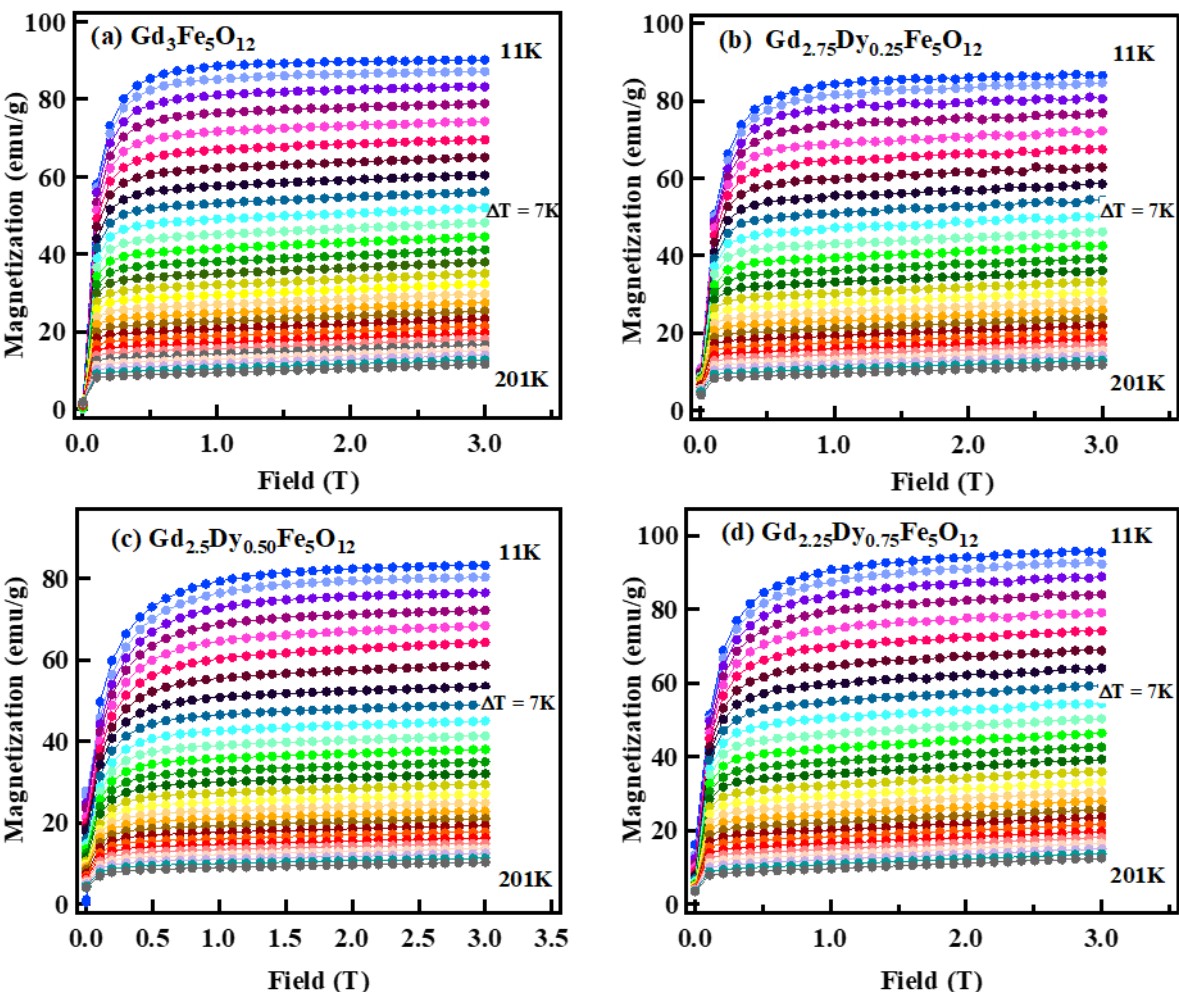

**Figure 20.** (**a**–**d**) Isothermal magnetization curves for the $Gd_{3-x}Dy_xFe_5O_{12}$ compound.

Figures 24–27 show the magnetic entropy change curve $-\Delta S_M(T)$ as a function of temperature for $Gd_{3-x}RE_xFe_5O_{12}$. The maxima ($-\Delta S_M^{max}$) for the $-\Delta S_M(T)$ curve is observed to be independent of temperature and field, as shown in Figure 28. As shown in Figure 24a–d, the $-\Delta S_M^{max}$ value for $Dy^{3+}$ doped garnet increases with $x$ content. The optimum value of magnetic entropy change reached 2.04 J.$Kg^{-1}K^{-1}$ for $x = 0.75$, which is ~7% higher than that of the $x = 0.0$ sample. An increase in magnetic entropy change with $Dy^{3+}$ substitution is due to the replacement of $Gd^{3+}$ (6.99 $\mu_B$) having a smaller magnetic moment by $Dy^{3+}$ (9.05 $\mu_B$) having a significant magnetic moment (from the DFT calculation above). The variation in $\Delta S_M$ is mainly due to the superexchange interaction between Fe–O–Fe ions.

By doping $Dy^{3+}$, the Dy-Fe superexchange interactions become strong, enhancing the $-\Delta S_M$ value [71]. The ($-\Delta S_M^{max}$) value decreases with $x$ content for the Nd, Sm, and Y doped garnet except for Sm ($x = 0.75$), as shown in Figures 25–27. The decreasing behavior of the magnetocaloric effect with $RE^{3+}$ doped garnet can be explained based on the magnetic moment of an individual element. From the theoretical observations, the magnetic moment of Nd ($-1.54$ $\mu_B$), Sm (2.55 $\mu_B$), and Y (0 $\mu_B$) are smaller than the magnetic moment of Gd (6.99 $\mu_B$). The $-\Delta S_M^{max}$ increases for $Sm^{3+}$ doped garnet for $x = 0.75$. The $\Delta S_M(T)$ plots show the broad curve covering a large temperature range with $RE^{3+}$ doped samples. Figure 28a–d shows the summary of the $-\Delta S_M^{max}$ value of $Gd_{3-x}RE_xFe_5O_{12}$ as a function of field. The maxima value ($-\Delta S_M^{max}$) shows the proportional relation with the applied field. The $-\Delta S_M^{max}$ values of some garnets are summarized in Table 10 to compare our results.

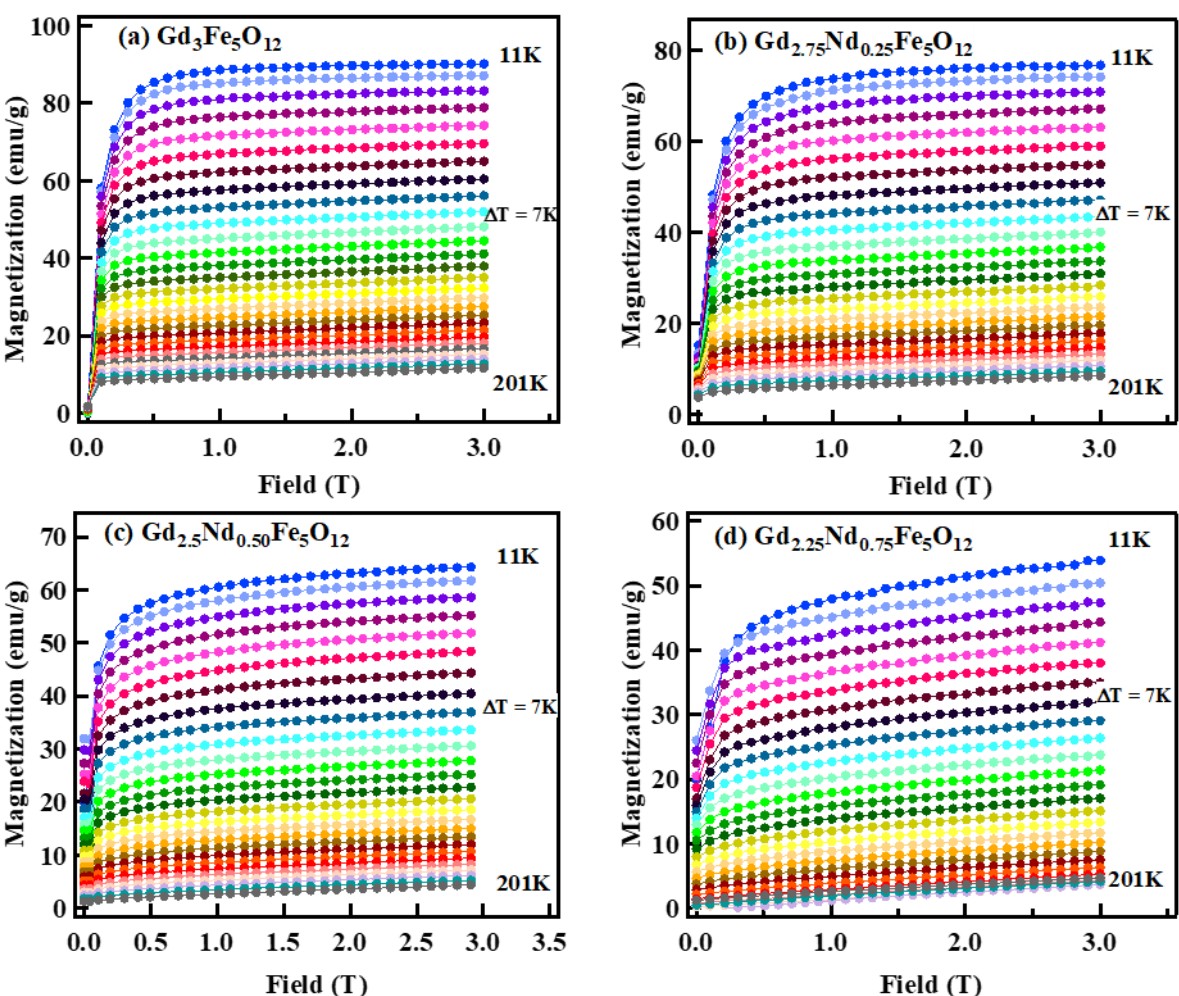

**Figure 21.** (**a**–**d**) Isothermal magnetization curves for the $Gd_{3-x}Nd_xFe_5O_{12}$ compound.

The relative cooling power (*RCP*) is a metric that quantifies the performance of magnetocaloric materials. The *RCP* value depends on the $-\Delta S_M$ and magnetocaloric materials' operating temperature range. The *RCP* is calculated as follows,

$$RCP = |\Delta S_M^{max}| \times \delta T_{FWHM} \tag{18}$$

where $\delta T_{FWHM}$ is the full-width-half-maxima obtained from the $-\Delta S_M$ vs. T plots of Figures 24–27.

Figure 29 shows the evolution of the *RCP* of $Gd_{3-x}RE_xFe_5O_{12}$ as a function of the applied magnetic field. The *RCP* value for $x = 0.0$ is ~31 J/kg at H = 0.5 T, which increases with the applied field and becomes ~219 J/kg at H = 3.0 T. The calculated *RCP* value for $Gd_{3-x}RE_xFe_5O_{12}$ is higher even at low fields than the other garnets reported in the literature [20,74,75]. The low field high *RCP* value of the $Gd_{3-x}RE_xFe_5O_{12}$ is very promising for the magnetic refrigeration application for low-temperature applications. The influence of the magnetic field on *RCP* may be estimated according to the formula,

$$RCP = A H^R \tag{19}$$

The R exponents obtained from the numerical fit of RCP are listed in Table 10. The *R*-value for $Gd_3Fe_5O_{12}$ is 1.10 and increases with $RE^{3+}$ substitution. The maximum *R*-value is obtained for the $Nd^{3+}$ (0.75) doped garnet. The *R*-value describes the field dependency of RCP. An *R*-value close to 1 implies the linear increase of RCP with the applied field.

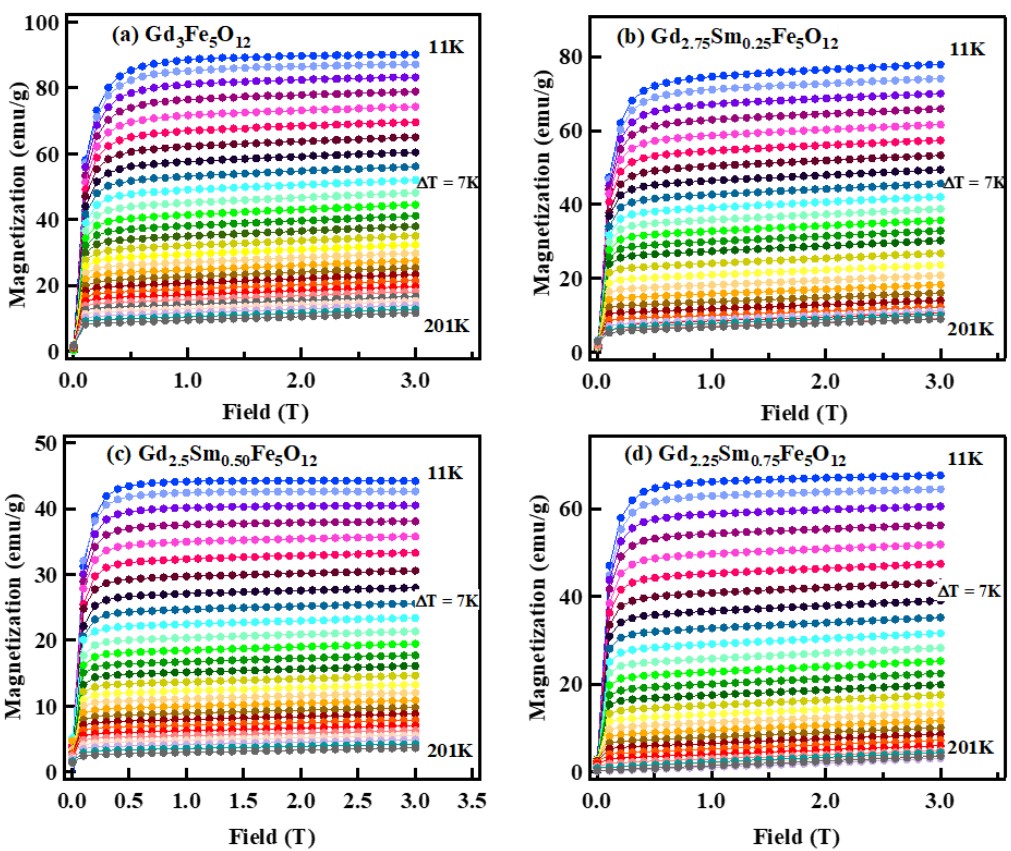

**Figure 22.** (**a**–**d**) Isothermal magnetization curves for the $Gd_{3-x}Sm_xFe_5O_{12}$ compound.

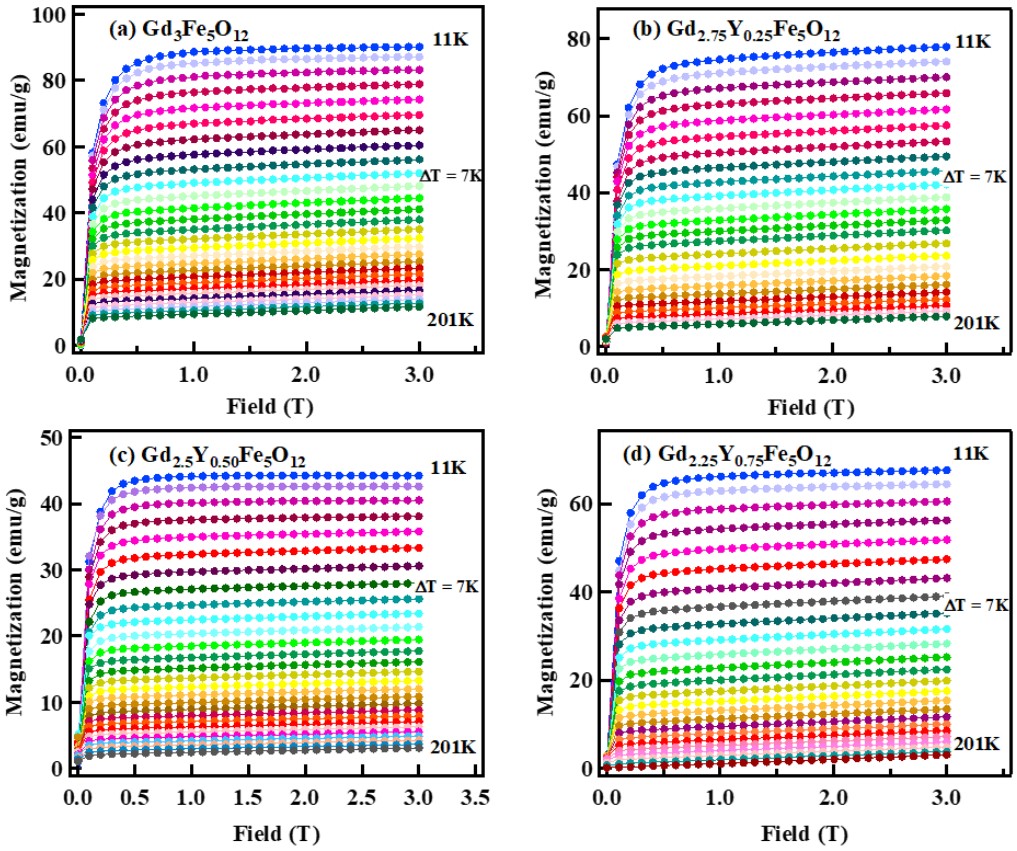

**Figure 23.** (**a**–**d**) Isothermal magnetization curves for the $Gd_{3-x}Y_xFe_5O_{12}$ compound.

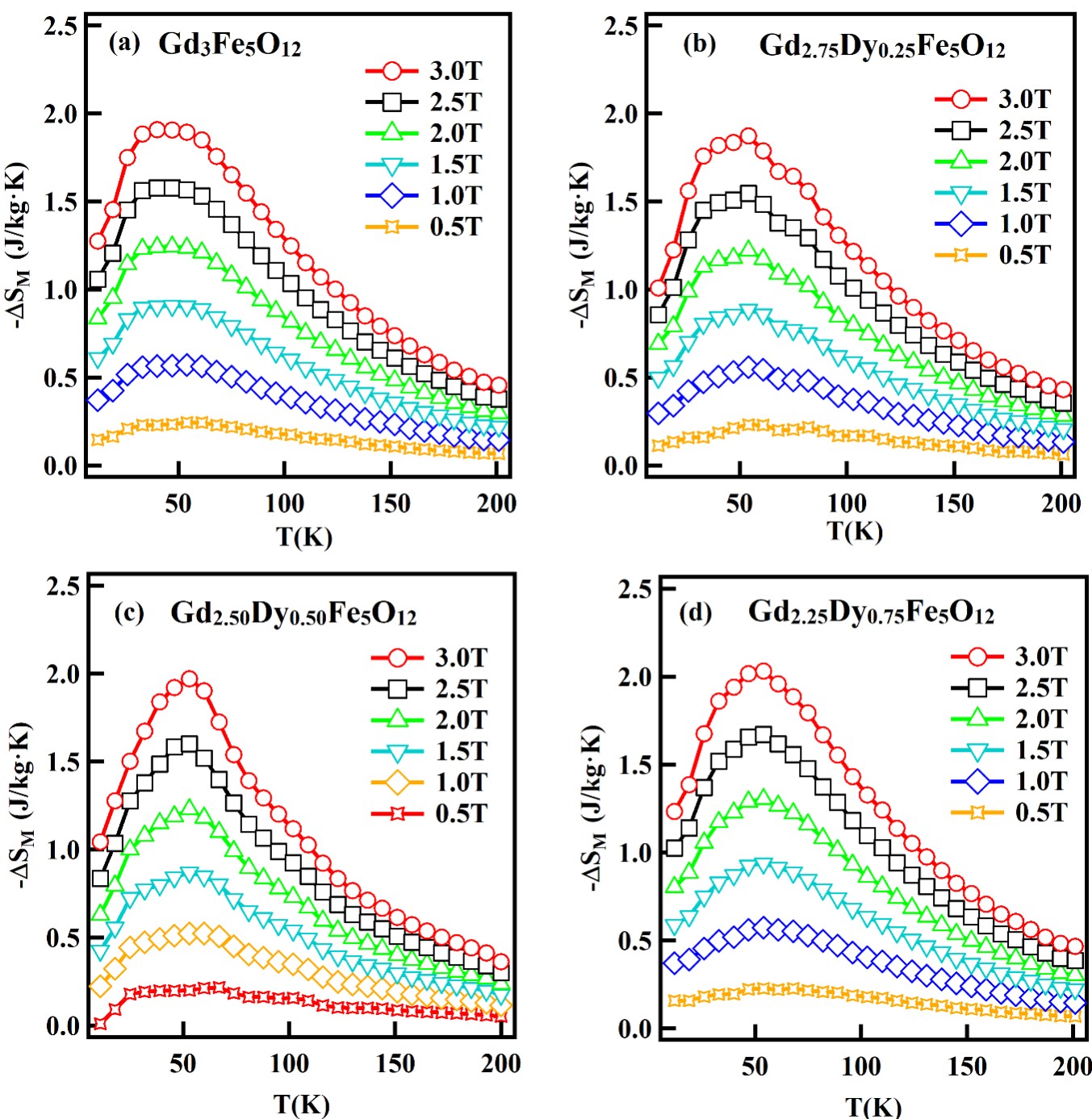

**Figure 24.** (**a**–**d**): Change in magnetic entropy $-\Delta S_M$, as a function of temperature up to 3 T fields for the $Gd_{3-x}Dy_xFe_5O_{12}$ compound.

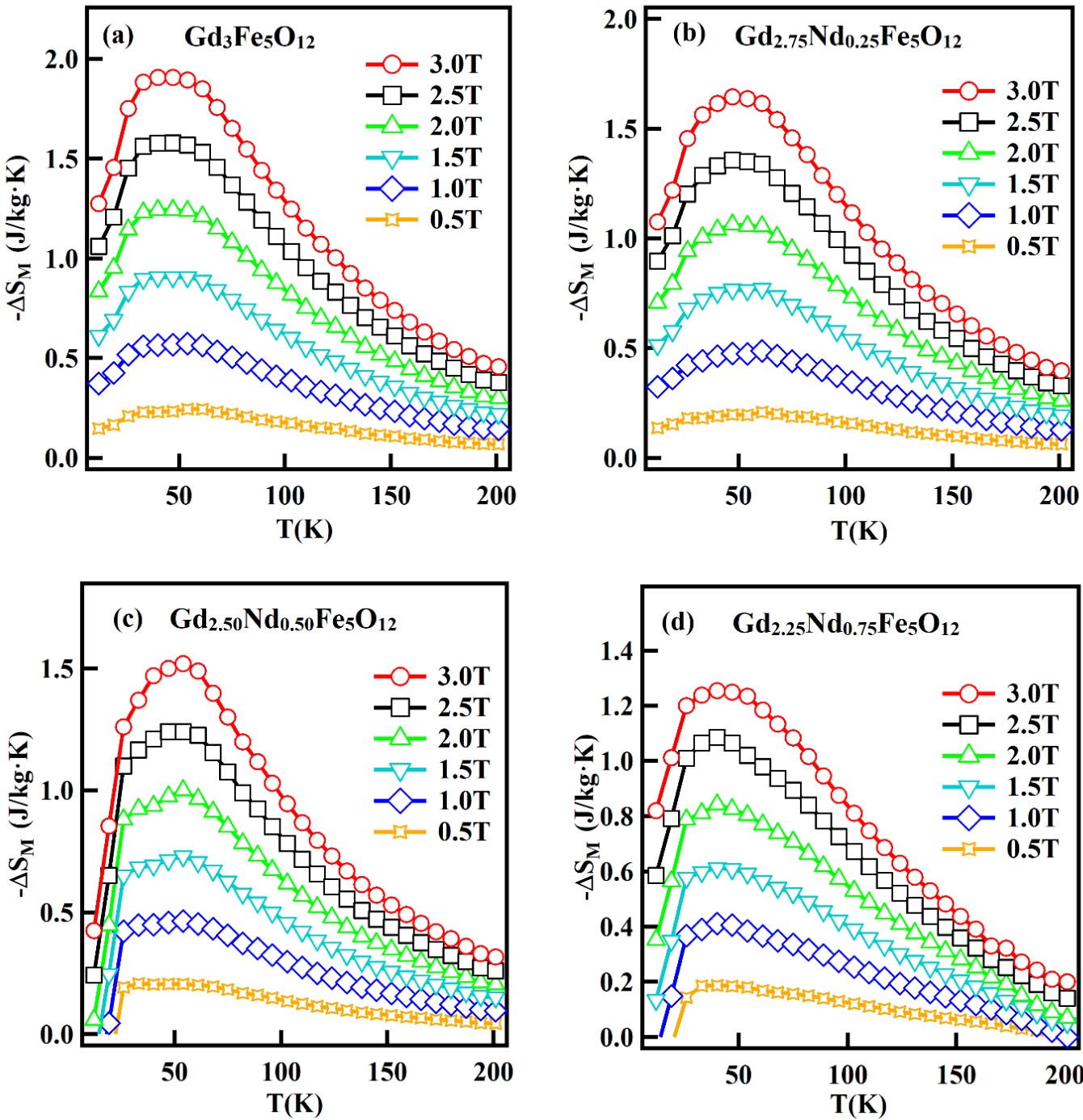

**Figure 25.** (**a**–**d**): Change in magnetic entropy $-\Delta S_M$, as a function of temperature up to 3 T fields for the $Gd_{3-x}Nd_xFe_5O_{12}$ compound.

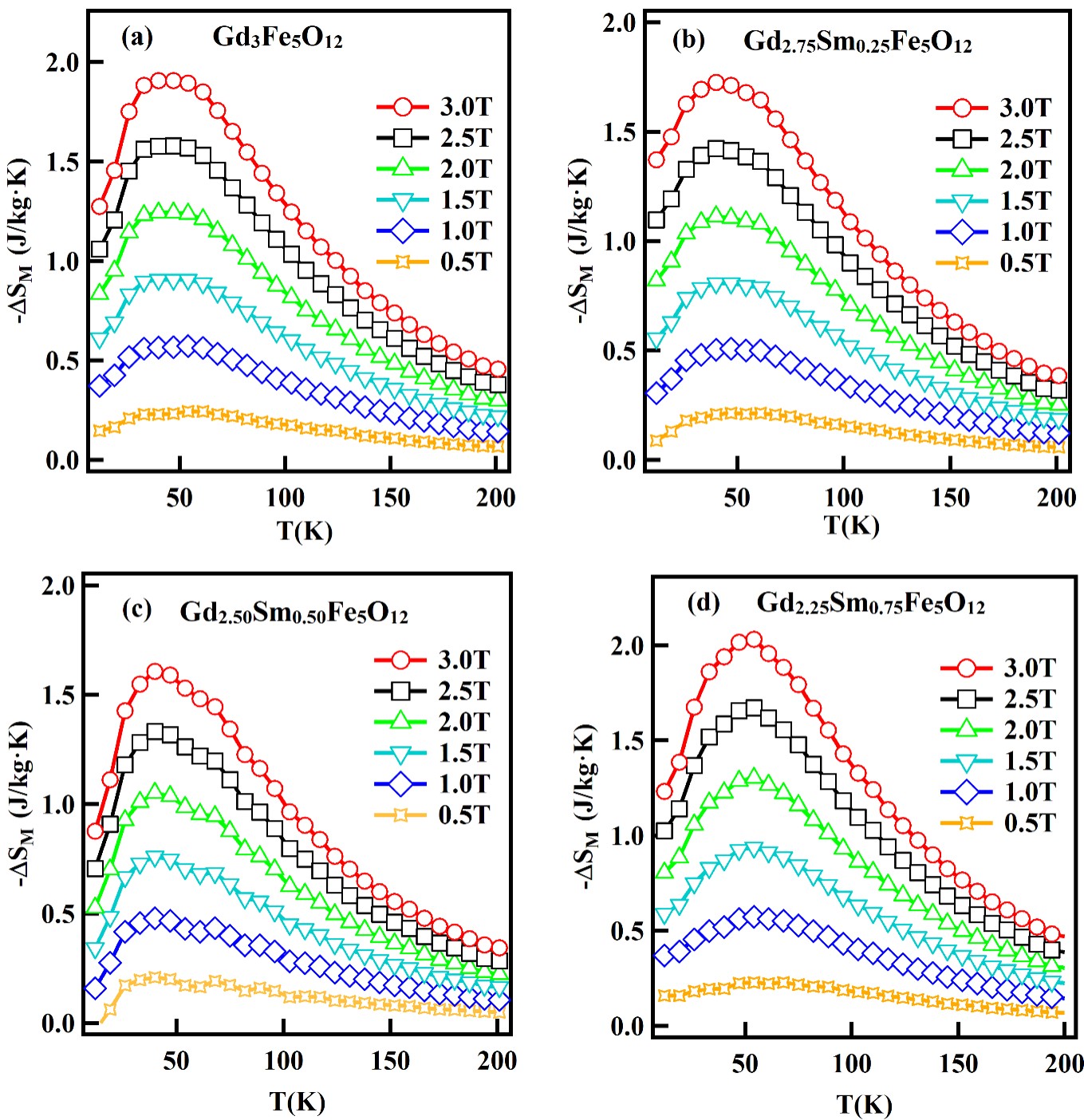

**Figure 26.** (**a**–**d**): Change in magnetic entropy $-\Delta S_M$, as a function of temperature up to 3 T fields for the $Gd_{3-x}Sm_xFe_5O_{12}$ compounds.

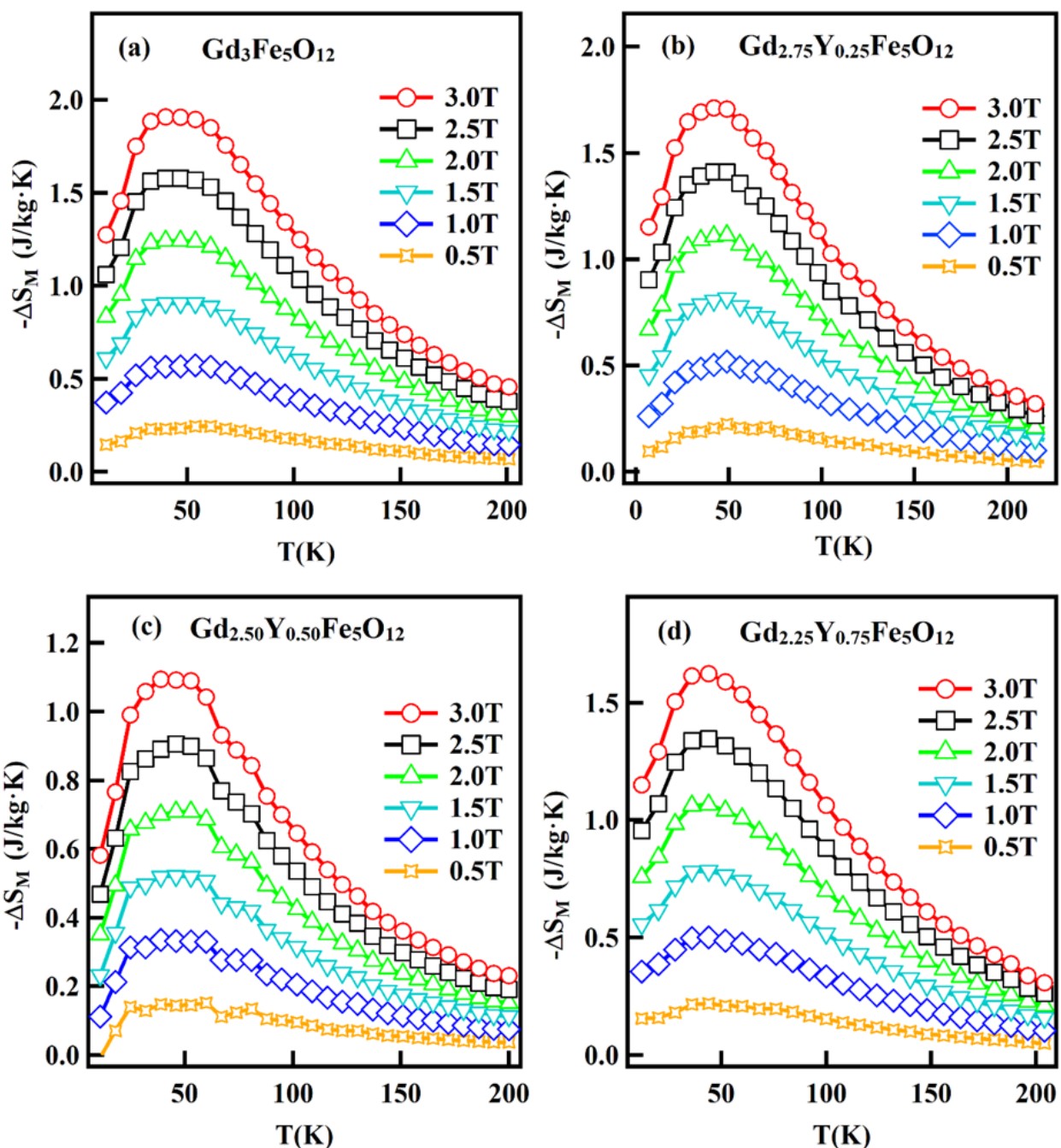

**Figure 27.** (**a**–**d**): Change in magnetic entropy $-\Delta S_M$, as a function of temperature up to 3 T fields for the $Gd_{3-x}Y_xFe_5O_{12}$ compound.

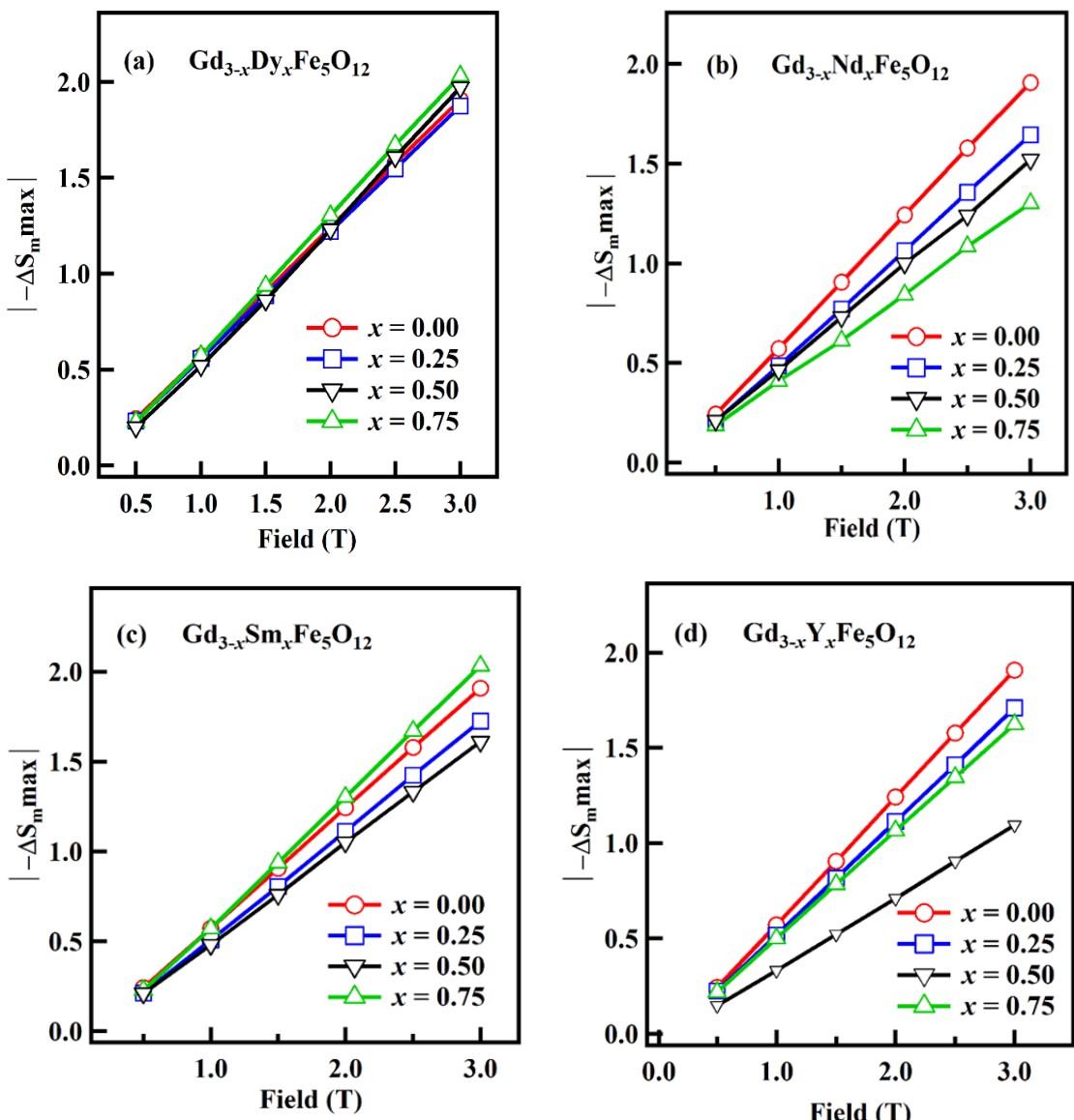

**Figure 28.** $\Delta S_M{}^{max}$ vs. field for the $Gd_{3-x}RE_xFe_5O_{12}$ compound.

**Table 10.** Comparison of the magnetocaloric parameters, $\Delta S_M{}^{max}$ and RCP, of a selection of garnets.

| Compound | $T_{peak}$ | $\Delta S_M{}^{max}$ | $H$ (T) | RCP | R | Reference |
|---|---|---|---|---|---|---|
| $Ho_3Fe_5O_{12}$ | 34 | 4.72 | 5 | 136 | | [16] |
| $Er_3Fe_5O_{12}$ | 24 | 4.94 | 5 | 103 | | [16] |
| $Gd_3Fe_5O_{12}$ | 40 | 1.99 | 3 | 193 | | [22] |
| $Dy_3Fe_5O_{12}$ | 58 | 2.03 | 3 | 165 | | [18] |
| $Gd_3Ga_{2.5}Fe_{2.5}O_{12}$ | 12 | ~1.70 | 1 | | | [72] |
| $Gd_{2.25}Dy_{0.75}Ga_{2.5}Fe_{2.5}O_{12}$ | 10 | ~1.55 | 1 | | | [18] |
| $Gd_3Fe_5O_{12}$ (bulk) | 40 | 0.45 | 1 | | | [73] |
| $Gd_3Fe_5O_{12}$ (50 nm) | 25 | 1.49 | 3 | | | [28] |
| $Gd_3Fe_5O_{12}$ | 40 | 1.91 | 3 | 219 | 1.10 | Present work |
| $Gd_{2.25}Dy_{0.75}Fe_5O_{12}$ | 54 | 2.04 | 3 | 234 | 1.13 | Present work |
| $Gd_{2.25}Nd_{0.75}Fe_5O_{12}$ | 40 | 1.25 | 3 | 140 | 1.14 | Present work |
| $Gd_{2.25}Sm_{0.75}Fe_5O_{12}$ | 54 | 2.03 | 3 | 234 | 1.13 | Present work |
| $Gd_{2.25}Y_{0.75}Fe_5O_{12}$ | 44 | 1.62 | 3 | 180 | 1.12 | Present work |

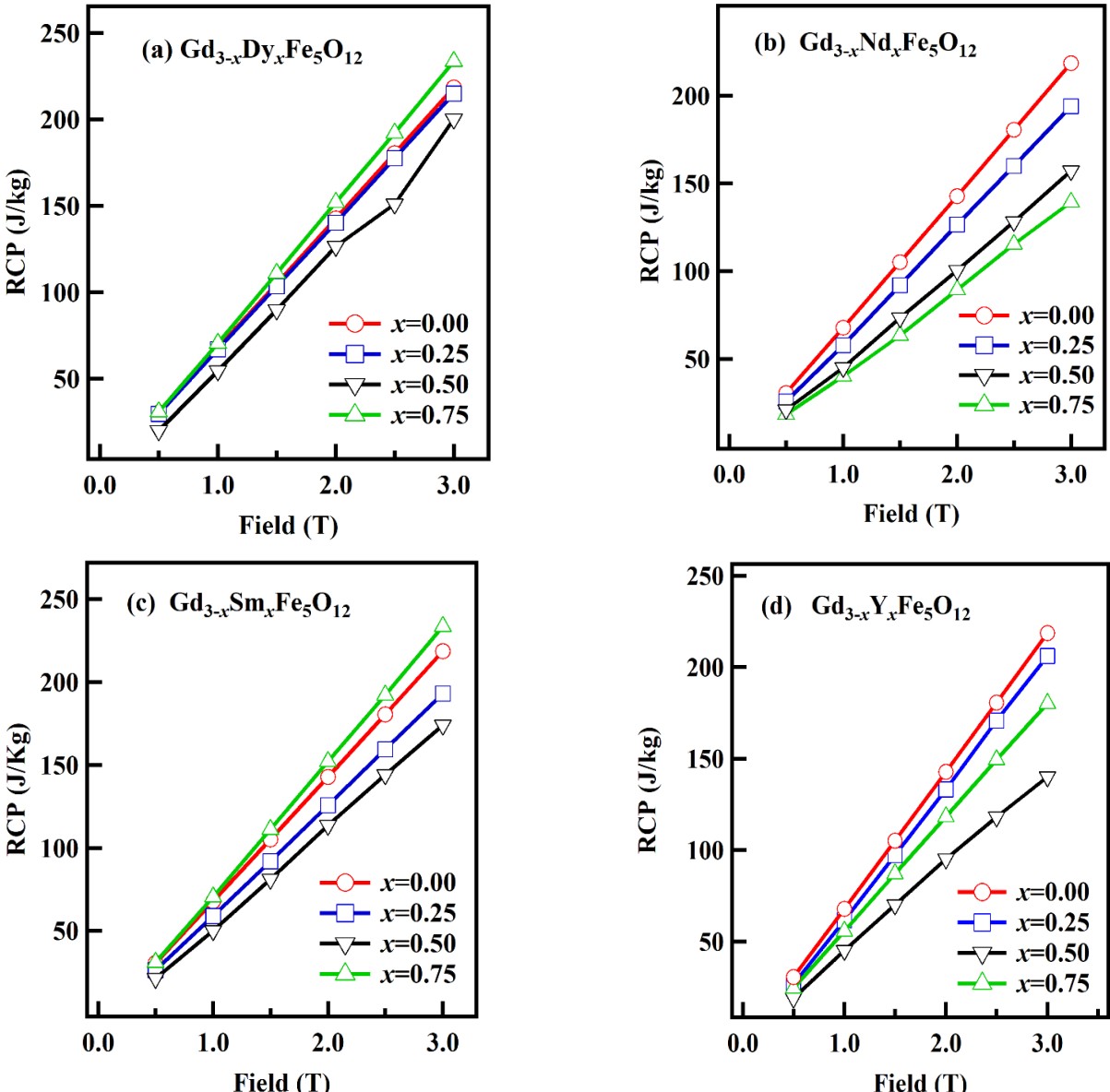

**Figure 29.** (**a–d**): Relative cooling power (RCP) of the $Gd_{3-x}RE_xFe_5O_{12}$ compound as a function of the applied field.

## 4. Conclusions

The synthesis of $RE^{3+}$ doped $Gd_{3-x}RE_xFe_5O_{12}$ ($x$ = 0.0, 0.25, 0.50, and 0.75, $RE^{3+}$ = Y, Nd, Sm, and Dy) was conducted successfully via the sol-gel autocombustion method. The substitution of $RE^{3+}$ ions on the $Gd^{3+}$ site of garnet brings in a structural and magnetic change in the compound. The XRD analysis shows the formation of a garnet structure with the Ia-3d space group. The Rietveld refinement shows that the lattice parameter decreased with $Dy^{3+}$ and $Y^{3+}$ substitution and increased with $Nd^{3+}$ and $Sm^{3+}$ substitution in accordance with the ionic radii of corresponding $RE^{3+}$ ionic radii. The bond angle between $RE^{3+}$-$O^{2-}$-$Fe^{3+}$ increased, $Fe(Oct.)^{3+}$–$O^{2-}$–$Fe(Tetra.)^{3+}$ decreased, and the bond length between $RE^{3+}$-$O^{-2}$ decreased with the $Dy^{3+}$ and $Y^{3+}$ doped sample. These structural changes have an essential influence on the magnetic structure of the compound. Magnetic studies reveal that the $Dy^{3+}$ substitution garnet shows higher saturation magnetization with a maximum value of 99 emu/g for $x$ = 0.75, whereas all other $RE^{3+}$ show a decrease in saturation magnetization value. The temperature-dependent magnetization study reveals that $RE^{3+}$ ions with non-zero magnetic moments couple strongly with the

Fe$^{3+}$(Tetra.) site. The Dy$^{3+}$ doped garnet shows the highest magnetic entropy change value compared to other RE$^{3+}$ doped garnets. The maxima value for Dy$^{3+}$ doped garnet achieved ($\Delta S_M{}^{max} \sim 2.00$ Jkg$^{-1}$K$^{-1}$) is due to the compound's sizeable magnetic moment. In summary, a substantial change in magnetic entropy value shows that rare-earth doped garnets could be suitable magnetocaloric materials for low-temperature cooling technology.

**Author Contributions:** Conceptualization, S.R.M.; methodology, S.R.M., D.N., A.K.P, N.K. and C.H.; software, X.S. and R.B.; validation, S.R.M. and A.K.P.; formal analysis, D.N.; investigation, D.N.; resources, S.R.M., A.K.P. and X.S.; data curation, D.N. and S.K.; writing—original draft preparation, D.N., S.R.M. and S.K.; writing—review and editing, S.R.M., S.K. and D.N.; visualization, S.R.M., A.K.P. and X.S.; supervision, S.R.M.; project administration, S.R.M.; funding acquisition, None. All authors have read and agreed to the published version of the manuscript.

**Funding:** This research received no external funding.

**Institutional Review Board Statement:** Not applicable.

**Informed Consent Statement:** Not applicable.

**Data Availability Statement:** The data presented in this study are available on request from the corresponding author.

**Acknowledgments:** Magnetic measurements were performed at the State University of New York (SUNY), Buffalo State, and supported by the National Science Foundation Award No. DMR-2213412. N.K. acknowledges financial support from the Office of Undergraduate Research, SUNY, Buffalo State. Computational resources were provided by the University of Memphis High-Performance Computing Center (HPCC).

**Conflicts of Interest:** The authors declare no conflict of interest.

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
