# Peer review of "Rare-Earth Doped Gd3−xRExFe5O12 (RE = Y, Nd, Sm, and Dy) Garnet: Structural, Magnetic, Magnetocaloric, and DFT Study"

_ceramics, doi:10.3390/ceramics6040120_

Round 1

Reviewer 1 Report

The article investigates the effect of rare-earth ion doping on the structural, magnetic and magnetocaloric properties of a garnet compound with potential applications in magnetic refrigeration. The authors present detailed information obtained by several different complementary theoretical and experimental methods for a comprehensive understanding of the structure-property relationship. The article contains a large amount of data presented in the form of tables and figures that illustrate the results well.

The article contains well-articulated objectives. The used research methods are relevant and solid. To obtain element specific magnetic data, an additional X-ray magnetic circular dichroism study in the future would be an advantage. The results presented in the article look reliable and accurate. The objectives have been achieved. The results presented in this article appear to be reliable and accurate. The article is well written and well structured.

A small remark: the sentence on page 11 line 252 is unclear or partially deleted.

Reviewer 2 Report

The paper contains several relevant pieces of information on the magnetic properties of rare-earth doped ceramic oxides. I recommend publication after revision and clarification of the following points:

1) It is reported that DFT+U was used for the electronic structure calculation. Did the authors also consider corrections for the exchange interactions (DFT+U+J)?

2) The authors should include more information about the geometry optimization: convergence criterion for the atomic forces and stress. Furthermore, it is not clear if the lattice was also optimized.

3) The authors should use their simulated geometry data to correlate with their initial discussion about lattice parameters, bond lengths and angles. Is there a good agreement between the simulated and experimental data?

4) I suggest that the authors include the values of the band gaps in their DFT discussion for the sake of completeness. Maybe a graph of the band gap as a function of the doping would be suitable for this section.

The text is very clear and easy to follow. However, the authors should double check typos and grammar one more time before publication.

Round 2

Reviewer 2 Report

After the clarifications, the present paper is suitable for publication.

No issues detected

Author Response

Thank you very much for your time and consideration.